# Active rock glaciers of the contiguous United States: GIS inventory and spatial distribution patterns

Gunnar Johnson[1], Heejun Chang[2], and Andrew Fountain[3]

[1]Environmental Science Department, Portland State University, Portland, Oregon, 97201, USA
[2]Geography Department, Portland State University, Portland, Oregon, 97201, USA
[3]Geology Department, Portland State University, Portland, Oregon, 97201, USA

*Correspondence to*: Gunnar Johnson (alpinebones@gmail.com)

**Abstract.** In this study we present the Portland State University Active Rock Glacier Inventory ($n$ = 10,332) for the contiguous United States, derived from the manual classification of remote sensing imagery (Johnson, 2020; https://doi.org/10.1594/PANGAEA.918585). Individually, these active rock glaciers are found across widely disparate montane environments, but their overall distribution unambiguously favors relatively high, arid mountain ranges with sparse vegetation. While at least one active rock glacier is identified in each of the 11 westernmost states, nearly 88% are found in just five states: Colorado ($n$ = 3889), Montana ($n$ = 1813), Idaho ($n$ = 1689), Wyoming ($n$ = 839), and Utah ($n$ = 834). Mean active rock glacier area is estimated at 0.10 km$^2$, with cumulative active rock glacier area totaling 1004.05 km$^2$. Active rock glaciers are assigned to a three-tier classification system based on area thresholds and surface characteristics known to correlate with downslope movement. Class 1 features ($n$ = 7042, average area = 0.12 km$^2$) appear to be highly active, Class 2 features ($n$ = 2415, average area = 0.05 km$^2$) appear to be intermediately active and Class 3 features ($n$ = 875, average area = 0.04 km$^2$) appear to be minimally active. This geospatial inventory will allow past active rock glacier research findings to be spatially extrapolated, help facilitate further active rock glacier research by identifying field study sites, and serve as a valuable training set for the development of automated rock glacier identification and classification methods applicable to other large regional studies.

## 1 Introduction

The most well-known elements of the alpine cryosphere are massive ice glaciers and perennial snowfields (simply "glaciers" and "snowfields" hereafter). Despite being among the most striking permafrost features, and likely due to their more nuanced definition and relatively difficult identification (Brardinoni et al., 2019), rock glaciers are a lesser known component of the alpine cryosphere. Though recent evidence shows that they are far more numerous than glaciers, they remain an under-studied and under-appreciated element of the cryosphere (Duguay et al., 2015). The spatial distributions of glaciers and snowfields of the contiguous U.S. are well understood (Fountain et al., 2017; RGI Consortium, 2017). Conversely, the distribution of rock glaciers of the contiguous U.S. is much less certain. Lacking the brilliantly reflective surfaces of glaciers and snowfields, which in late summer afford strong spectral contrast with immediately adjacent land

cover, rock glaciers are challenging to identify remotely using automated methods, making spatial inventories difficult to compile (Millar and Westfall, 2008). The widely accepted continuum concept places rock glaciers somewhere between glaciers, which are composed almost completely of ice and have a low mineral content, and creeping permafrost, which is composed almost completely of mineral fractions and has a low ice content (Haeberli et al., 2006; Berthling, 2011; Anderson

et al., 2018).  Virtually all rock glaciers form in cryo-conditioned landscapes, resulting from precipitation, meltwater or groundwater percolating into mechanically weathered debris and subsequently freezing (Francou et al., 1999; Berthling, 2011). This interstitial ice is shielded from direct solar insolation and insulated from warm air temperatures during the melt season by the overlying regolith mantle (Jones et al., 2019a). Provided some fraction of the internal ice content remains frozen through the summer, additional ice is incorporated each winter until a rock glacier is formed. Most researchers

consider active rock glaciers, the focus of this study, to be flowing bodies of permafrost, composed of generally regular vertical distributions of coarse talus and granular regolith bound by interstitial ice (Clark et al., 1998, Berthling and Etzelmuller, 2011). In this regard we agree with the active rock glacier definition, "… lobate or tongue-shaped bodies of perennially frozen unconsolidated material supersaturated with interstitial ice and ice lenses that move downslope or downvalley by creep as a consequence of the ice contained in them and which are, thus, features of cohesive flow", proposed

by Barsch (1996).

Rock glaciers that are not actively flowing are commonly classified as inactive, fossil, or relict rock glaciers, and were deliberately excluded from this inventory due to their difficult identification through manual classification of aerial imagery.

Rock glaciers often cease to flow due to severely reduced fractions, and in many cases a near total absence, of interstitial ice. Additionally, rock glaciers can also cease to flow when the topographic gradients they rest on become too shallow, as in the bottom of a cirque, or when debris supply is constrained. This means that active and inactive rock glaciers are often found colocated, at similar elevations, and experiencing similar climatic conditions.  While we do not mean to discount the climatological research interest of inactive rock glaciers, confidently identifying them though remote sensing imagery

analysis alone is exceptionally difficult, and results from any such attempts should be further investigated by detailed and direct geophysical field examination (Colucci et al., 2019). In many cases inactive rock glaciers ceased flowing hundreds or thousands of years ago, allowing widespread alpine soil and vegetation community development on their surfaces. Indeed, recent research has shown that when attempting to discriminate active rock glaciers from inactive rock glaciers, surficial vegetation cover is the most statistically significant predictor (Kofler et al., 2020).  Additionally, these soils and vegetation

readily obscure most of the visual evidence of their past activity readily identifiable through remote sensing image analysis, and as such inactive rock glaciers were intentionally excluded from this active rock glacier inventory due to severe limitations in our ability to confidently identify them based on the methods and data sets employed. However, this active rock glacier inventory can readily and directly be compared to major components of other rock glacier inventories, provided those inventories clearly identify which features are active and which features are inactive. Furthermore, previous rock

glacier inventories that have attempted to identify both active and inactive rock glaciers have generally found the two feature types are often colocated, meaning the active rock glacier inventory presented here will be a useful starting point for any future efforts to inventory inactive rock glaciers of the contiguous United States.

Debris-covered glaciers are a landform closely related to active rock glaciers that most researchers have generally defined to essentially be talus-covered alpine glaciers, retaining discrete ice cores with relatively low internal concentrations of regolith (Berthling, 2011). The surficial talus mantling of debris-covered glaciers is generally sourced from mass wasting of over-steepened lateral slopes, often formerly buttressed by the glacier body, but now unsupported and exposed to the elements due to glacial recession. In most cases, fully mantled debris-covered glaciers with thick and continuous surficial debris layers are

virtually indistinguishable from the more traditionally defined active rock glaciers through surface analysis alone, either in the field or based on remote sensing imagery. Generally, fully mantled debris covered glaciers with thick and continuous surficial debris layers can only be confidently identified by direct coring or ground penetrating radar, though debris-covered glaciers with expansive surfaces of exposed ice in their accumulation zones and/or thin and discontinuous surficial debris layers are readily discriminated from active rock glaciers through remote sensing imagery analysis. Additionally, in cases

where supraglacial lakes and/or streams are present on the surfaces of debris-covered glaciers, these features can be used to discriminate them from active rock glaciers. The nuances of classifying these two cryospheric feature types (e.g., internal ice fraction thresholds, contiguity and extent of ice cores, etc.) is occasionally debated, but is not an issue we seek to resolve with this inventory (Potter, 1972; Clark et al., 1998; Haeberli et al., 2006; Berthling, 2011). While we have made every effort to exclude debris-covered glaciers from this inventory (Fig.1), our methods cannot completely discriminate between fully

mantled debris-covered glaciers that lack expansive surfaces of exposed ice in their accumulation zones or obvious supraglacial lakes/streams and traditionally defined active rock-glaciers. Regardless, virtually all examples of both fully mantled debris-covered glaciers that lack expansive surfaces of exposed ice in their accumulation zones or obvious supraglacial lakes/or streams and traditionally defined active rock glaciers have been shaped by a combination of glacial and periglacial forces at some point in their geologically recent history. Indeed, there is considerable evidence that, especially in

a rapidly warming world, debris-covered glaciers often transition into active rock glaciers (Anderson et al., 2018; Jones et al., 2019a). As such, we believe any inadvertent inclusion of fully mantled debris-covered glaciers that lack expansive surfaces of exposed ice in their accumulation zones or obvious supraglacial lakes/streams in this active rock glacier inventory should not dramatically impair the utility of the inventory in furthering understanding of the alpine cryosphere.

In this study we develop and present the Portland State University Active Rock Glacier Inventory (PSUARGI) for the contiguous United States (Johnson, 2020). This inventory will help further define the role of active rock glaciers with respect to alpine climatology, ecology, geomorphology, hydrology, and engineering. Rock glacier responses to climate shifts are beginning to be understood with equal specificity to the climatic responses of glaciers, allowing past climatic conditions on

short (Bodin et al., 2009; Sorg et al., 2015) and long time scales (Konrad et al., 1999; Stenni et al., 2007; Matthews et al., 2013) to be inferred from their present condition and distribution. The PSUARGI will also help advance growing ecological interest in rock glaciers as climate refugia for cold-adapted flora and fauna (Brighenti et al., 2021; Caccianiga et al., 2011; Harrington et al., 2017; Hayashi, 2020; Sulejman, 2011; Millar et al., 2013b). Previously studied active rock glaciers have shown they can control major fractions of local regolith transport (Kaab and Reichmuth, 2005; Haeberli et al., 2006). Rock glaciers have also been shown to have considerable water storage capacities, and are important modulators of surface runoff, especially in arid alpine environments where they are present (Halla et al., 2021). Additionally, and especially when compared to glaciers, rock glacier meltwaters exhibit unique hydrographs (Bajewsky and Gardner, 1989; Jones et al., 2019b) and hydrochemistry signatures (Millar et al., 2013a; Fegel et al., 2016), as well as also volumetric discharge increases in late summer due to climate change (Caine, 2010). From an anthropogenic perspective, active rock glaciers represent unique engineering challenges, particularly with regard to the possibility of catastrophic collapse and debris flow generation (Iribarren and Bodin, 2010; Lugon and Stoffel ,2010; Bodin et al., 2017), but they also offer engineering opportunities as reservoirs of construction aggregate and water (Burger et al., 1999).

The regional or continental scale impacts of these and other rock glacier influences identified in previous research on individual active rock glaciers cannot be inferred without an accurate active rock glacier inventory at the same spatial scale. Smaller scale rock glacier inventories have been completed before (Table 1), but the active rock glacier distribution across an area the size of the contiguous U.S. has never been quantified in a comprehensive manner. Where prior rock glacier inventories considered study areas most often measured in dozens, hundreds, or, occasionally, thousands of square kilometers, our active rock glacier inventory evaluates a study area of over 3,000,000 km$^2$. This study addresses a pressing research question: What is the spatial distribution of active rock glaciers of the contiguous U.S.?

## 2 Data and Methods

### 2.1 Study Region and Data Sources

We used Google Earth Pro 7.1.7 (Google Earth, 2018) and ESRI ArcMap 10.4 software (ESRI, 2017) to search for active rock glaciers. Google Earth Pro provides imagery acquired at multiple dates from the early 1990s to present, orthorectified to accurate and easily manipulated three-dimensional surfaces. Quick access to multiple images of the same location, captured at different times of day, during different seasons, and across multiple years facilitated active rock glacier identification certainty. We relied on Google Earth Pro and the three-dimensional elevation models it provides for most identifications, supplementing with National Agricultural Imagery Program (NAIP, 2019) plan-view imagery imported into ArcMap 10.4 when Google Earth Pro imagery was unsuitable due to cloud cover, snow cover, or other issues.

We initially began evaluating all montane regions of the contiguous U.S., but failed to find any evidence of active rock glaciers east of the Rocky Mountain States. Therefore, we focused our efforts on the 11 westernmost states (Arizona (AZ), California (CA), Colorado (CO), Idaho (ID), Montana (MT), New Mexico (NM), Nevada (NV), Oregon (OR), Utah (UT), Washington (WA), Wyoming (WY)). Climatologically, this study region is defined by four zones of the NOAA U.S. Climate Region system (Karl and Koss 1984): the Northwest Climate Region (hereafter "NW Region") of ID, OR and WA;

the Southwest Climate Region (hereafter "SW Region") of AZ, CO, NM and UT; the West Climate Region (hereafter "W Region") of CA and NV; and the West North Central Climate Region (hereafter "WNC Region") of MT and WY. The major mountain range in each of the four Regions is the Cascades, Southern Rockies, Sierra Nevada and Northern Rockies, respectively.

**2.2 Active Rock Glacier Identification**

Because glaciers, snowfields, and active rock glaciers are often co-located (Jones et al., 2019a; Knight et al., 2019; Millar and Westfall, 2019), we used two GIS inventories that identify relevant features to inform target areas for our initial search for active rock glaciers; the Randolph Glacier Inventory (RGI) v6.0 (Fountain et al., 2017; RGI Consortium, 2017) and the National Land Cover Database (NLCD) 2011 (Homer et al., 2015). The RGI is focused only on glaciers, whereas the NLCD identifies any perennial snow or ice feature. From this initial effort and our growing expertise in locating active rock

glaciers, we expanded our search areas to explore alpine regions far from any inventoried glaciers or perennial snow or ice features, but that could potentially host active rock glaciers.

Active rock glaciers were identified manually by their distinct surface characteristics (Aoyama, 2005; Haeberli et al., 2006). These characteristics include ridge and swale surface banding resulting from differential flow rates and terminal and lateral

slopes over-steepened beyond the angle of repose, presumably cemented by interstitial ice. Common mass wasting processes responsible for individual fragments of regolith traveling downslope result in accumulations at or below the angle of repose. Similar approaches to active rock glacier identification, focusing on surface topography characteristics identified from aerial and satellite imagery, have been applied in other previous research (Eztelmuller et al., 2007; Janke, 2007; Degenhardt, 2009; Janke et al., 2015; Millar et al., 2019).


We focused our inventory efforts on identifying active rock glaciers that, surficially, appear to contain appreciable internal ice fractions and are presently or were recently flowing downslope. We follow previous studies that omit features with expansive bare glacial ice in their accumulation zones or obvious supraglacial lakes/streams as those are clearly debris-covered glaciers, but make no further attempt to discriminate active rock glaciers from fully mantled debris-covered glaciers

(Bodin et al., 2010; Berthling 2011, Perucca and Angillieri, 2011). After the exponentially larger study area than any previously investigated, a second major distinction between our active rock glacier inventory and classification system and other previous U.S. rock glacier inventory efforts is that we intentionally attempt to exclude inactive rock glaciers. We

ignored potential candidate features lacking over-steepened terminal slopes and/or present evidence of advanced surficial soil development, such as expansive vegetation growth, both of which imply the rock glacier has a small internal ice fraction and has not flowed downslope recently.

When identifying a candidate active rock glacier, plan-view images were initially viewed at 1:2000 scale or better. Once suspected ridge and swale flow banding and over-steepened terminal and lateral slopes were identified, image scale was greatly increased. All available clear sky images of the same scene were then evaluated, with plan views being replaced by oblique views from multiple angles and multiple scales and three-dimensional topography exaggerated by 50%. The perimeter of individual active rock glaciers were manually delineated using Google Earth Pro. Usually, sharp changes in slope were evident, indicating a perimeter boundary between the thickened ice-bound regolith of the active rock glacier and the surrounding unconsolidated talus of the adjacent slope. Additionally, lower active rock glacier margins often abut well-vegetated terrain. The upper margins are often defined by a change in slope, from the steep slopes of exposed bedrock and unconsolidated talus in the rock glacier accumulation zone to the more gentle slope of the main body of the ice-thickened active rock glacier. Generally, active rock glacier boundary confidence is highest along sharp terminal and lateral margins and lowest along accumulation zones where exposed bedrock is not present. When considering multi-lobate active rock glaciers we focused on distinct accumulation zones to ascribe individual lobes to a given active rock glacier. While every effort was made to apply these guidelines consistently, we readily concede that identifying and delineating rock glaciers remotely is technically challenging and subject to individual interpretation and best professional judgement. Past evaluation of remote rock glacier inventory methods has shown high degrees of variability between even well trained image analysts, particularly with regard to rooting zones (Brardinoni et al., 2019), and we support ongoing efforts to standardize methods for rock glacier inventories within the research community.

Understandably, there can be some disagreement between analysts regarding rock glacier classification (Brardinoni et al., 2019). To partially address this ambiguity all features identified as active rock glaciers were subsequently assigned to a three-tier classification system based on surface characteristics known to correlate with downslope movement motivated by deformation of the internal ice-rock matrix (Fig. 2), particularly the presence and extent of ridge and swale flow banding (Haeberli et al., 2006; Brenning et al., 2012; Liu et al., 2013). Class 1 rock glaciers appear to be highly active, exhibit unambiguous, complex and extensive ridge and swale flow banding, and have substantially over-steepened terminal and lateral boundaries. Class 2 rock glaciers appear to be intermediately active, exhibit some pronounced ridge and swale flow banding, and have somewhat over-steepened terminal and lateral boundaries. Class 3 rock glaciers appear to be minimally active, exhibit sparse ridge and swale flow banding, and have intermittently over-steepened terminal and lateral boundaries.

To characterize the topographic characteristics of the individual active rock glaciers identified, elevation data were extracted from the USGS National Elevation Dataset (NED) ⅓ arc-second (≈ 10 m) digital elevation model (USGS, 2017).

Topographic variables of elevation, slope, aspect, and insolation were determined using Spatial Analyst tools in ArcMap 10.4 (ESRI, 2017). Active rock glacier area was calculated in km$^2$, while slope and aspect were calculated in degrees. Aspect was decomposed to an eastness and northness component (Nussear et al., 2009), and solar insolation was calculated in watt-hours per m$^2$. To characterize the climate of the active rock glaciers, climate data, including air temperature and precipitation, were also extracted from PRISM 1981 - 2010 climate normals (PRISM, 2017) using Spatial Analyst tools in ArcMap 10.4. PRISM data were also used to calculate several derivative atmospheric variables, such as fraction of precipitation falling as snow and mean vapor pressure deficit, using the Raster Calculator tool in ArcMap 10.4. These publicly available climate data have a spatial resolution of 800 m, with an average daily accumulated total precipitation bias of less than 2.5% in the western US, 1961 - 2001 (DiLuzio et al., 2008). Active rock glacier classification and area clustering analysis using Moran's I-statistics helped further describe active rock glacier spatial distributions (Cliff and Ord, 1971; Senn, 1976; TiefelsdorfTiefelsdorf, 2002).

## 3 Results

### 3.1 Overall Distribution

We identified 10,332 active rock glaciers (Class 1 = 7042, Class 2 = 2415, Class 3 = 875) across the western U.S. (Fig. 3, Table 2), after removing 146 small (< 0.01 km$^2$) Class 3 rock glaciers following glaciological convention of area thresholds (Navarro and Magnusson 2017). This minimum area threshold was also selected due to decreased confidence in extremely small rock glacier identification, as well as an attempt to ensure all features included in the inventory were active rock glaciers exhibiting downslope movement modulated by internal deformation of ice, something that would be exceedingly rare in any rock glaciers smaller than 0.01 km$^2$. Average active rock glacier area is 0.10 km$^2$ and the average distance between each active rock glacier and its nearest neighbor is 0.69 km. Contiguous U.S. active rock glaciers have an average elevation of 3144.3 m, an average slope of 20.51°, an average eastness of -0.007, and an average northness of 0.066 (Fig.4). Climatically, the average annual active rock glacier precipitation is 350.2 mm, the average air temperature is 0.19 °C, the average dew point temperature is -8.37 °C, and the average vapor pressure deficit is 4.52 hPa (Fig. 4). Differences were noted in rock glacier topographic and climatic attributes between NOAA Climate Regions (Fig. 5). The overall active rock glacier centroid (41.5332,-110.7083) is located in the southwest corner of the WNC Region (Fig. 3). The centroids of each of the three active rock glacier classes (Class 1 = (41.5112, -110.5556), Class 2 = (41.7012, -111.0141), Class 3 = (41.2470, -111.0942)) can be contained by a minimum bounding area circle with a diameter of 57.7 km. Moran's I analysis shows active rock glacier classifications and areas are significantly clustered (Table 3 and Table 4).

### 3.1.1 Regional Distributions

In the NW Region, we identified 1993 active rock glaciers (Class 1 = 1293, Class 2 = 512, Class 3 = 188)(Fig. 6). Geographically, the average active rock glacier size is 0.07 km$^2$, and the average distance between each active rock glacier and its nearest neighbor is 0.99 km. Topographically, the average active rock glacier elevation is 2629.6 m, the average slope is 20.7º, the average eastness is 0.000, and the average northness is 0.109 (Fig. 5).Climatically, the average annual active rock glacier precipitation is 365.4 mm, the average air temperature is 1.06 °C, the average dew point temperature is -7.47°C, and the average vapor pressure deficit is 4.85 hPa (Fig. 5). The NW Region active rock glacier centroid (44.8620, -115.2736) is located in the Sawtooth Mountains of Idaho (Fig. 3). The NW Region centroids of each of the three active rock glacier classes (Class 1 = (44.7208, -114.9471), Class 2 = (45.0615, -115.7468), Class 3 = (45.2899, -116.2301)) can be contained by a minimum bounding area circle with a diameter of 106.3 km (Fig. 6).

In the SW Region, we identified 4870 active rock glaciers (Class 1 = 3291, Class 2 = 1133, Class 3 = 446)(Fig. 7). The average SW Region active rock glacier size is 0.09 km$^2$, and the average distance between each SW Region active rock glacier and its nearest neighbor is 0.59 km. Topographically, the average active rock glacier elevation is 3490.35 m, the average slope is 20.70º, the average eastness is -0.013, and the average northness is 0.046 (Fig. 5).Climatically, the average annual active rock glacier precipitation is 335.12 mm, the average air temperature is -0.09 °C, the average dew point temperature is -8.92 °C, and the average vapor pressure deficit is 4.50 hPa (Fig. 5). The SW Region active rock glacier centroid (38.9385, -107.3569) is located in the Rocky Mountains of Colorado (Fig. 3). The SW Region centroids of each of the three active rock glacier classes (Class 1 = (38.9066, -107.2755), Class 2 = (39.0867, -107.5456), Class 3 = (38.7968, -107.4786)) can be contained by a minimum bounding area circle with a diameter of 38.2 km (Fig. 7).

In the W Region, we identified 817 active rock glaciers (Class 1 = 552, Class 2 = 181, Class 3 = 84)(Fig. 8). The average W Region active rock glacier size is 0.12 km$^2$, and the average distance between each W Region active rock glacier and its nearest neighbor is 0.68 km. Topographically, the average active rock glacier elevation is 3412.2 m, the average slope is 20.9º, the average eastness is -0.001, and the average northness is 0.082 (Fig. 5).Climatically, the average annual active rock glacier precipitation is 367.79 mm, the average air temperature is 0.61 °C, the average dew point temperature is -9.52 °C, and the average vapor pressure deficit is 5.07 hPa (Fig. 5). The W Region active rock glacier centroid (37.5421, -118.6340) is located in the Sierra Nevada of California (Fig. 3). The W Region centroids of each of the three active rock glacier classes (Class 1 = (37.5506, -118.6616), Class 2 = (37.4045, -118.6486), Class 3 = (37.7828, -118.4209)) can be contained by a minimum bounding area circle with a diameter of 48.0 km (Fig. 8).

In the WNC Region, we identified 2652 active rock glaciers (Class 1 = 1906, Class 2 = 589, Class 3 = 157)(Fig. 9). The average WNC Region active rock glacier size is 0.11 km$^2$, and the average distance between each WNC Region active rock

glacier and its nearest neighbor is 0.79 km. Topographically, the average active rock glacier elevation is 2813.0 m, the average slope is 19.9º, the average eastness is -0.002, and the average northness is 0.067 (Fig. 5). Climatically, the average annual active rock glacier precipitation is 361.2 mm, the average air temperature is -0.07 °C, the average dew point temperature is -7.7 °C, and the average vapor pressure deficit is 4.13 hPa (Fig. 5). The WNC Region active rock glacier centroid (45.0260,-110.9904) is located in the Rocky Mountains of Montana (Fig. 3). The WNC Region centroids of each of the three active rock glacier classes (Class 1 = (44.9782, -110.8925), Class 2 = (45.1292, -111.2260), Class 3 = (45.2200, -111.2951)) can be contained by a minimum bounding area circle with a diameter of 41.5 km (Fig. 9).

## 4 Discussion

### 4.1 Spatial Distribution Patterns

Individually, contiguous U.S. active rock glaciers are found across widely disparate montane environments, but their overall distribution unambiguously favors relatively high, arid mountain ranges with sparse vegetation. Active rock glacier populations in those regions are denser, and the individual active rock glaciers making up those populations are larger and exhibit surficial evidence of higher activity, than those of active rock glaciers found in humid mountain ranges with copious vegetation. Active rock glaciers of the NW Region are largest and most densely concentrated in the Sawtooth Mountains of Idaho. Active rock glaciers of the SW Region are largest and most densely concentrated in the Front Range and San Juan Mountains of Colorado and the Uinta Mountains of Utah. Active rock glaciers of the W Region are largest and most densely concentrated in the Sierra Nevada of California. Active rock glaciers of the WNC Region are largest and most densely concentrated in the Beartooth Mountains of Montana and the Absaroka Range of Wyoming.

### 4.2 Inventory Accuracy

The completeness and accuracy of the active rock glacier inventory were qualitatively and quantitatively supported by numerous field observations and remote sensing classification verification by multiple GIS analysts familiar with the alpine cryosphere generally and rock glaciers specifically. The lead author personally visited more than 50 active rock glaciers during field campaigns for related research, and more than 150 individual active rock glaciers with precise coordinates listed in past peer reviewed research were examined remotely when developing our classification criteria. While developing the inventory, dozens of test areas measuring 500 km² or greater in all 11 western states were checked by two other well trained GIS analysts familiar with the alpine cryosphere for "missing" active rock glaciers not originally identified by the lead author, and none were found. When considering the three-class active rock glacier activity classification scheme, a test subset of 60 randomly selected active rock glaciers were classified in isolation using the qualitative classification rules previously described by five GIS analysts familiar with the alpine cryosphere generally and rock glaciers specifically. Individual analyst classifications were then compared using Tukey's HSD test ($\alpha = 0.05$), yielding no significant differences

between analyst interpretations. Class 1 rock glaciers showed a 92% agreement between analysts, Class 2 rock glaciers an 87% agreement between analysts, and Class 3 rock glaciers a 79% agreement between analysts.

As this active rock glacier inventory is of unprecedented spatial extent, no analogous previous inventories exist for us to make direct and detailed GIS comparisons to over the entire study region. While smaller regional-scale U.S. rock glacier inventories have been compiled in the past, none of these inventories are publicly available as geospatial data sets. Coarse scale comparisons, however, were completed based on reported findings and figures published in previous studies presenting the aforementioned smaller regional U.S. rock glacier inventories. To compare our active rock glacier inventory and previous regional U.S. rock glacier inventories we created polygons using the corner coordinates of low resolution regional study maps from peer-reviewed articles highlighting one Colorado rock glacier inventory (Janke, 2007) and two California rock glacier inventories (Millar and Westfall, 2008; Liu et al., 2013). Polygons representing the extents of maps from the smaller regional inventories were then used to select simple counts of active rock glaciers identified in our inventory and compare them to counts of rock glaciers reported in the aforementioned studies. The 2007 Colorado inventory reported 28 "active" rock glaciers, the category in that study defined most similarly to our Class 1 classification criteria, in and around Rocky Mountain National Park, while we identified 29 Class 1 rock glaciers in the same region. The 2008 California study reported 184 rock glaciers in the central Sierra Nevada, but used a more inclusive "rock-ice feature" definition that deliberately includes inactive rock glaciers, than our active rock glacier classification criteria, while we identified 116 active rock glaciers of any class in the same region. The 2013 California study (Liu et al., 2013) reported 67 "active" rock glaciers, a subset of features identified in the 2008 study and the category in that study most similar to our Class 1 classification criteria, while we identified 88 active rock glaciers in largely the same study region. These three comparisons, and the agreement between the aforementioned inventories and our findings, greatly bolster our confidence in the overall accuracy of the PSUARGI.

**4.3 Inventory Applications**

Though our classification system and deliberate omission of inactive rock glaciers due to limitations in the analysis techniques (Brardinoni et al., 2019) and data sets available will undoubtedly preclude some desired applications of this active rock glacier inventory such as validating permafrost extent models (Boeckli et al., 2012; Schmid et al., 2015), we believe it represents an import step towards a fuller understanding of rock glaciers of the contiguous U.S. regardless. Several potential uses of this active rock glacier inventory are readily apparent, and we hope all will be explored by the research community in due time. Most immediately, this inventory will allow rapid identification of potential field sites for researchers interested in direct study of individual rock glaciers. Many researchers likely do not appreciate just how close their universities or labs already are to active rock glaciers, and this inventory would also offer powerful insights for any researchers eager to inventory inactive rock glaciers. Water resource managers in the arid western U.S. should also take note

of active rock glaciers, as the sizes and locations of these features are likely to play an increasingly important role in changing water supplies (Wagner et al., 2020a; Wagner et al., 2020b). Finally, we hope this inventory will aid ongoing refinement and future implementation of truly automated rock glacier detection methods. The ability to quickly, accurately and objectively identify rock glaciers from presently available remote sensing imagery, without relying on skilled visual image analysts or needing to address the inevitable interpretation disagreements between those analysts, would be an invaluable tool for climatologists, ecologists and many others (Brenning, 2009).

## 5 Data Availability

The PSUARGI geospatial data (Johnson, 2020) is available online via the PANGAEA data repository at https://doi.org/10.1594/PANGAEA.918585

## 6 Conclusions

We present an active rock glacier inventory much larger in both spatial extent and feature count than any previously completed in the U.S., covering a study area of over 3,000,000 km$^2$ and identifying 10,332 active rock glaciers. The densest active rock glacier distributions are found in mountain ranges that host no glaciers and very few snowfields, such as the Sawtooth Mountains of Idaho and the Uinta Mountains of Utah. Active rock glaciers are ubiquitous across wide swaths of the contiguous U.S. not often acknowledged by policy makers and water resource managers as being part of the alpine cryosphere, and their climatological, ecological and hydrologic importance cannot be underestimated. In the majority of regions of the contiguous U.S. where high, arid peaks well above treeline are found, active rock glaciers are found as well. While this inventory is in no way intended to be the final word on active rock glacier distributions of the contiguous U.S., we believe it will be valuable tool in future research aimed at better understanding the influence of climate change on these areas.

in the world to date, a powerful tool informing a wide range of research and management applications. The PSURGI exposes, for the first time at such an expansive spatial scale, what an ubiquitous component of the alpine cryosphere rock glaciers truly are. Despite their ubiquity, rock glaciers remain an under-studied and under-appreciated element of the alpine cryosphere (Duguay et al. 2015). The deeper understanding of where rock glaciers form and persist where provided by this inventory will aid ongoing refinement and future implementation of truly automated rock glacier detection methods. The ability to quickly, accurately and objectively identify rock glaciers from presently available remote sensing imagery, without relying on skilled visual image analysts or needing to address the inevitable interpretation disagreements between those

analysts, would be an invaluable tool for climatologists, ecologists, water resource managers and many others (Brenning 2009).

**7 Author Contributions**

Gunnar Johnson designed the research project, created and analyzed the active rock glacier inventory data, and wrote the manuscript. Heejun Chang and Andrew Fountain designed the research project and edited the manuscript

**8 Author Contributions**

The authors declare that they have no conflict of interest.

**9 Acknowledgements**

Kristina Dick, Kelly Hughes, Michelle Neeson, Justin Ohlschlager, and Matthias Weislogel all assisted in verifying active rock glacier classifications.

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

**Figures**

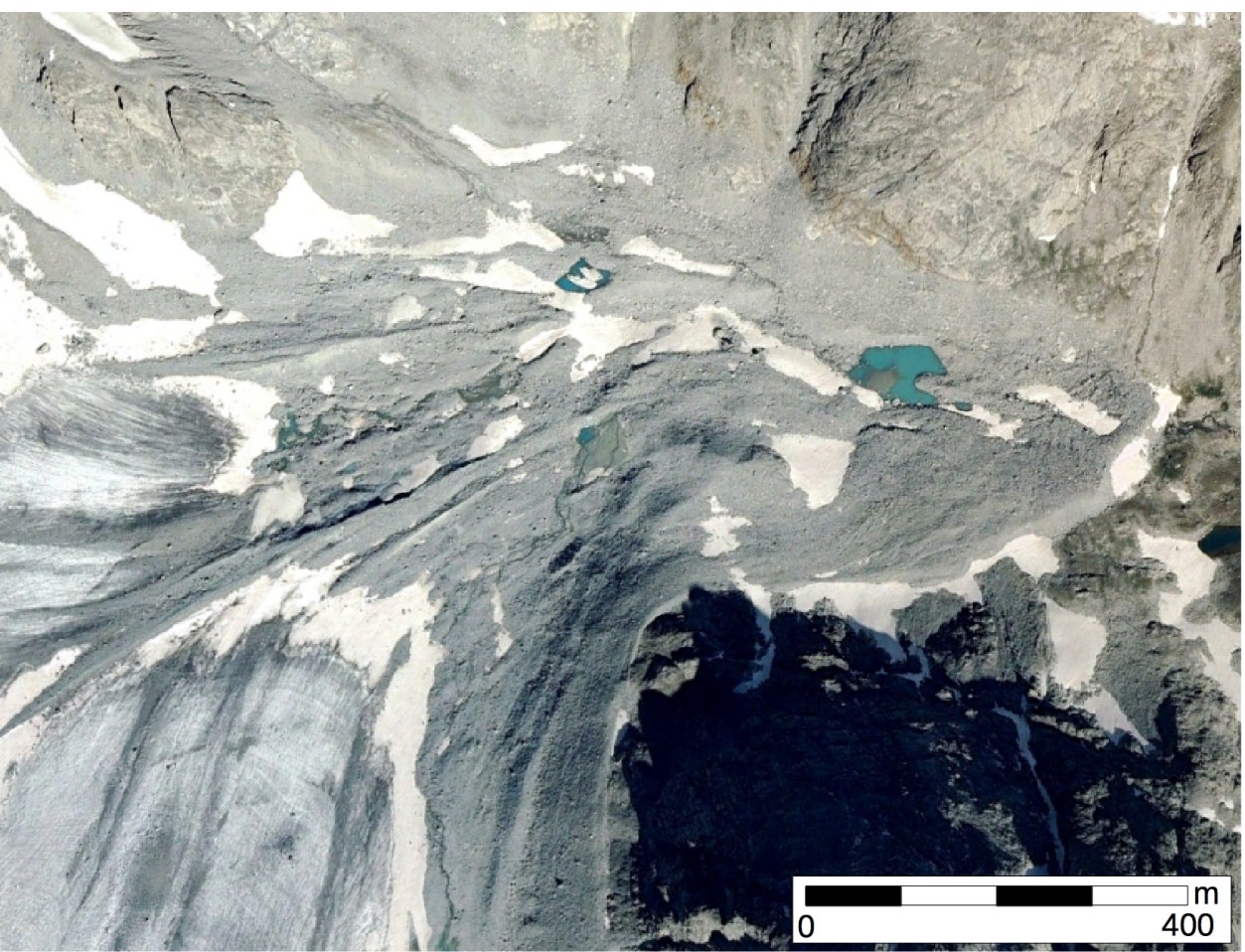

**Figure 1: Example of a prototypical debris-covered glacier, exhibiting expansive surfaces of exposed ice in the accumulation zone and obvious supraglacial lakes and streams on its surface. This example typifies the debris-covered glacier features we deliberately set out to exclude from this inventory. Image credit: ©Google Earth/Copernicus.**


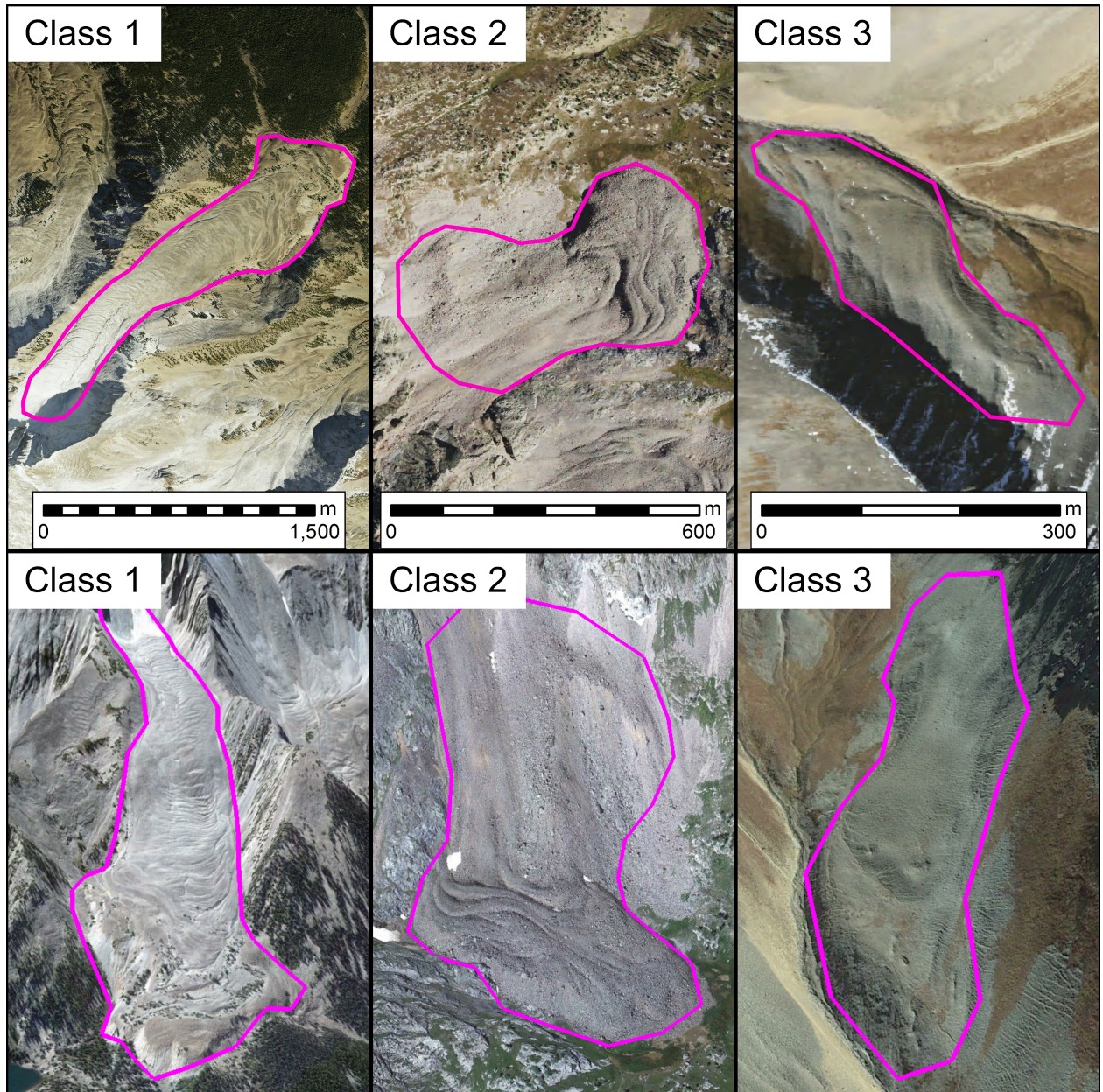

Figure 2: Examples of each of the three rock glacier classes shown in both plan view (top panels) and oblique upslope view (bottom panels). Leftmost panels show a Class 1 rock glacier (appears to be highly active, exhibits unambiguous, complex and extensive ridge and swale flow banding, and has substantially over-steepened terminal and lateral boundaries). Center panels show a Class 2 rock glacier (appears to be intermediately active, exhibits some pronounced ridge and swale flow banding, and has somewhat over-steepened terminal and lateral boundaries.). Rightmost panels show a Class 3 rock glacier (appears to be minimally active, exhibits sparse ridge and swale flow banding, and has intermittently over-steepened terminal and lateral boundaries.). Note different scale bars for each plan view panel, and that scale varies across images in oblique view panels. Image credit: ©Google Earth/Copernicus.

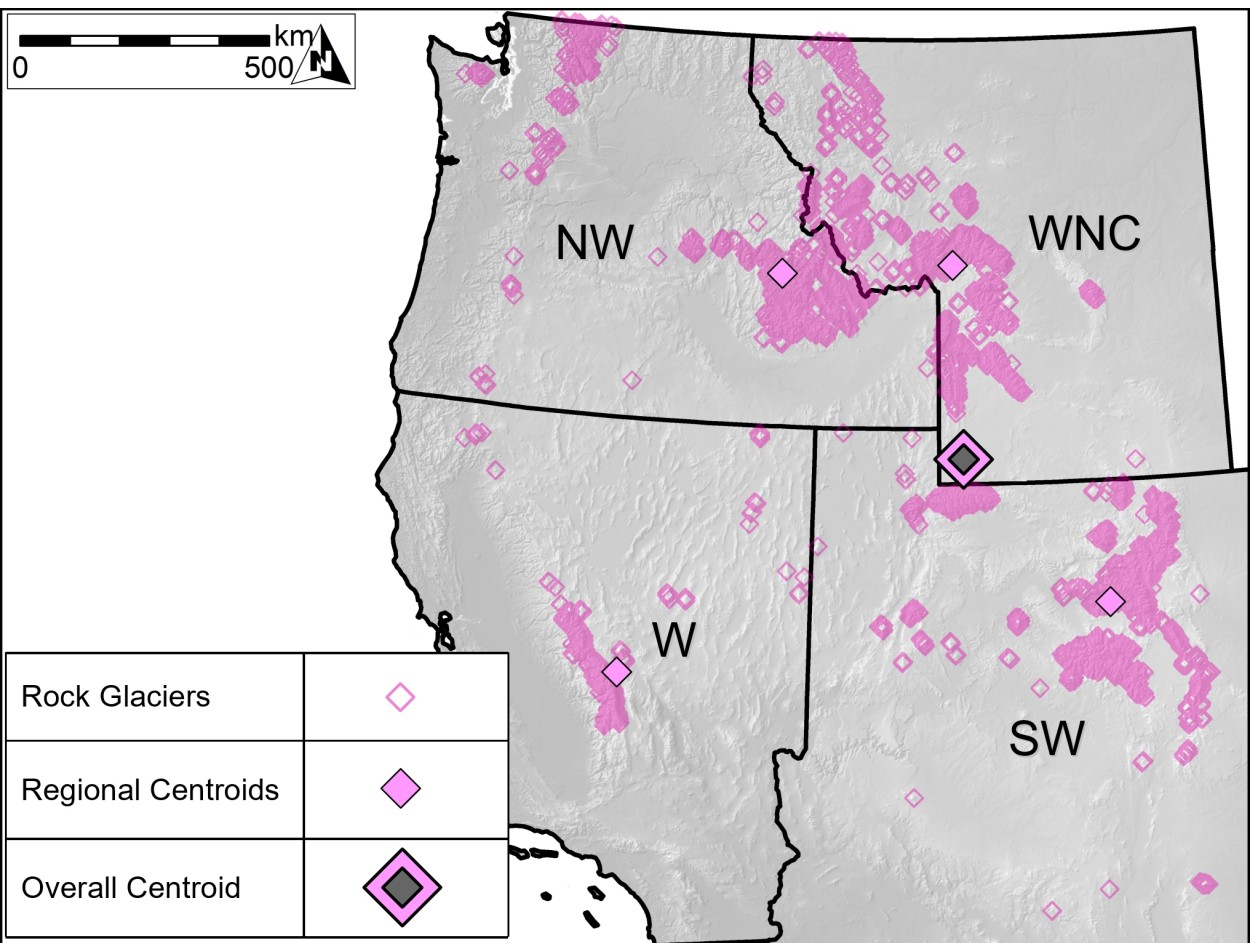

**Figure 3: Locations of rock glacier inventory features (n = 10,332), as well as centroids for the entire inventory and NOAA Climate Region subsets. The largest rock glaciers, as well as highest rock glacier densities, are found in the relatively arid Southern Rocky Mountains. The Sierra Nevada of California and Uinta Mountains of Utah, climatologically similar to the Southern Rockies, also host large rock glaciers at high densities. Rock glaciers of the humid Cascade Mountains are smaller and less densely distributed, and only a few pockets of rock glaciers are found south of 35° N latitude. However, the western U.S. is generally defined by mountainous, high elevation terrain, and rock glaciers are found in all 11 western states.**




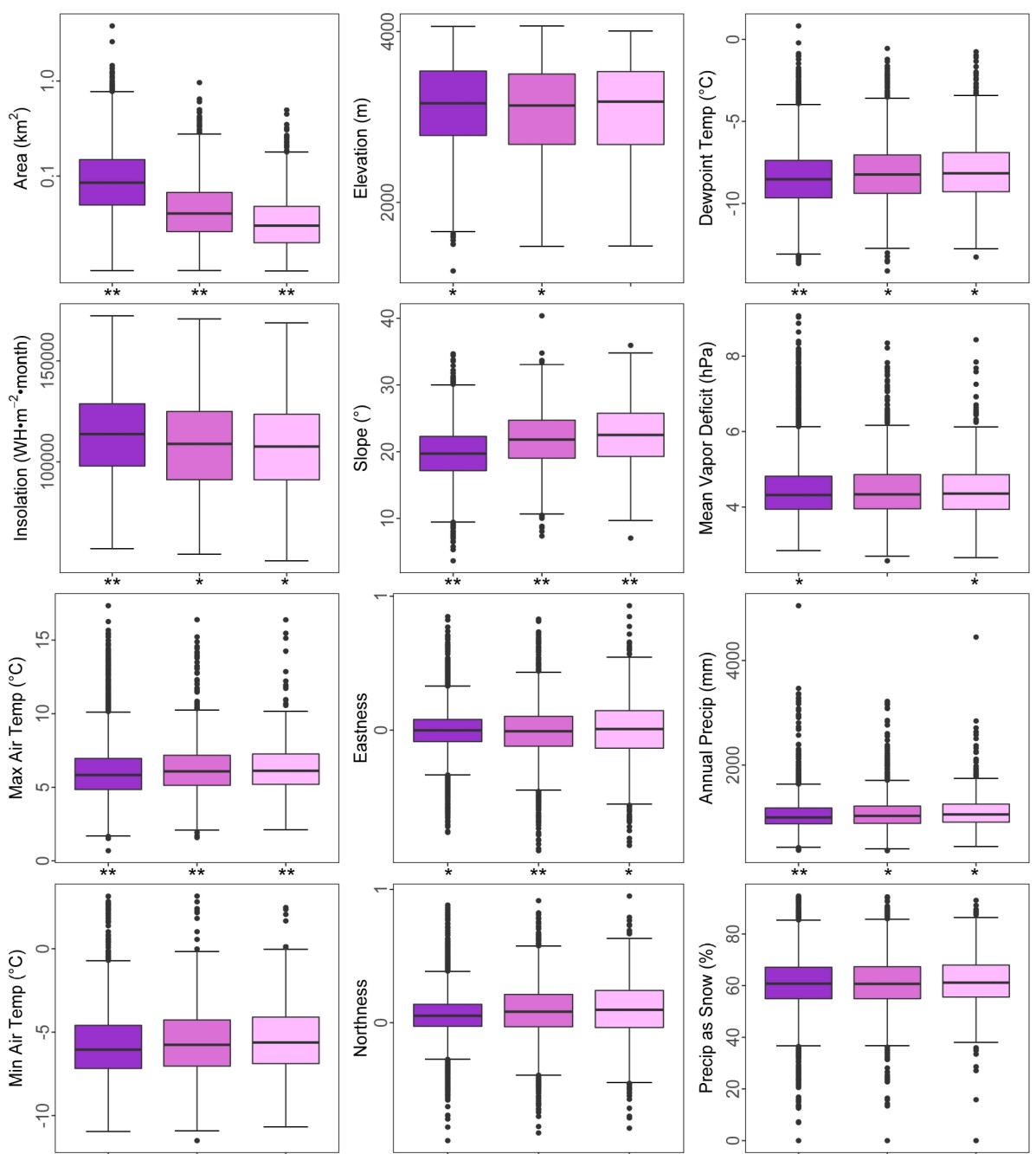

Figure 4: Geographic characteristics of Class 1 (dark purple, n = 7042), Class 2 (magenta, n = 2415) and Class 3 (light pink, n = 875) rock glaciers. Statistically significant differences (Tukey's HSD test, α = 0.05) are denoted with asterisks (different from one = *, different from both = **). Boxplot whiskers represent 1.5 times the interquartile range, outliers beyond those values are shown by solid dots.

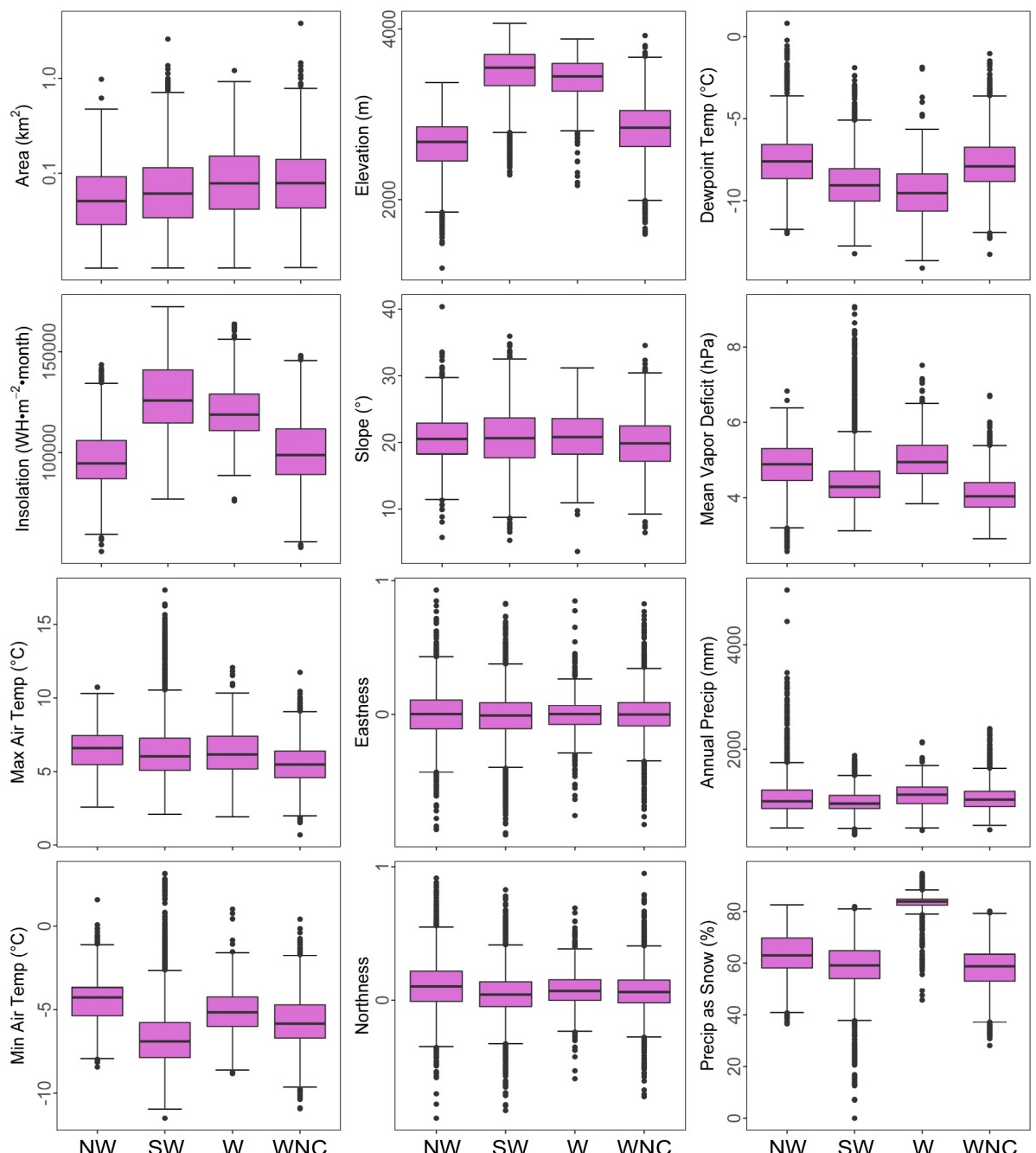

**Figure 5: Geographic characteristics of rock glaciers by NOAA Climate Region. Boxplot whiskers represent 1.5 times the interquartile range, outliers beyond those values are shown by solid dots.**


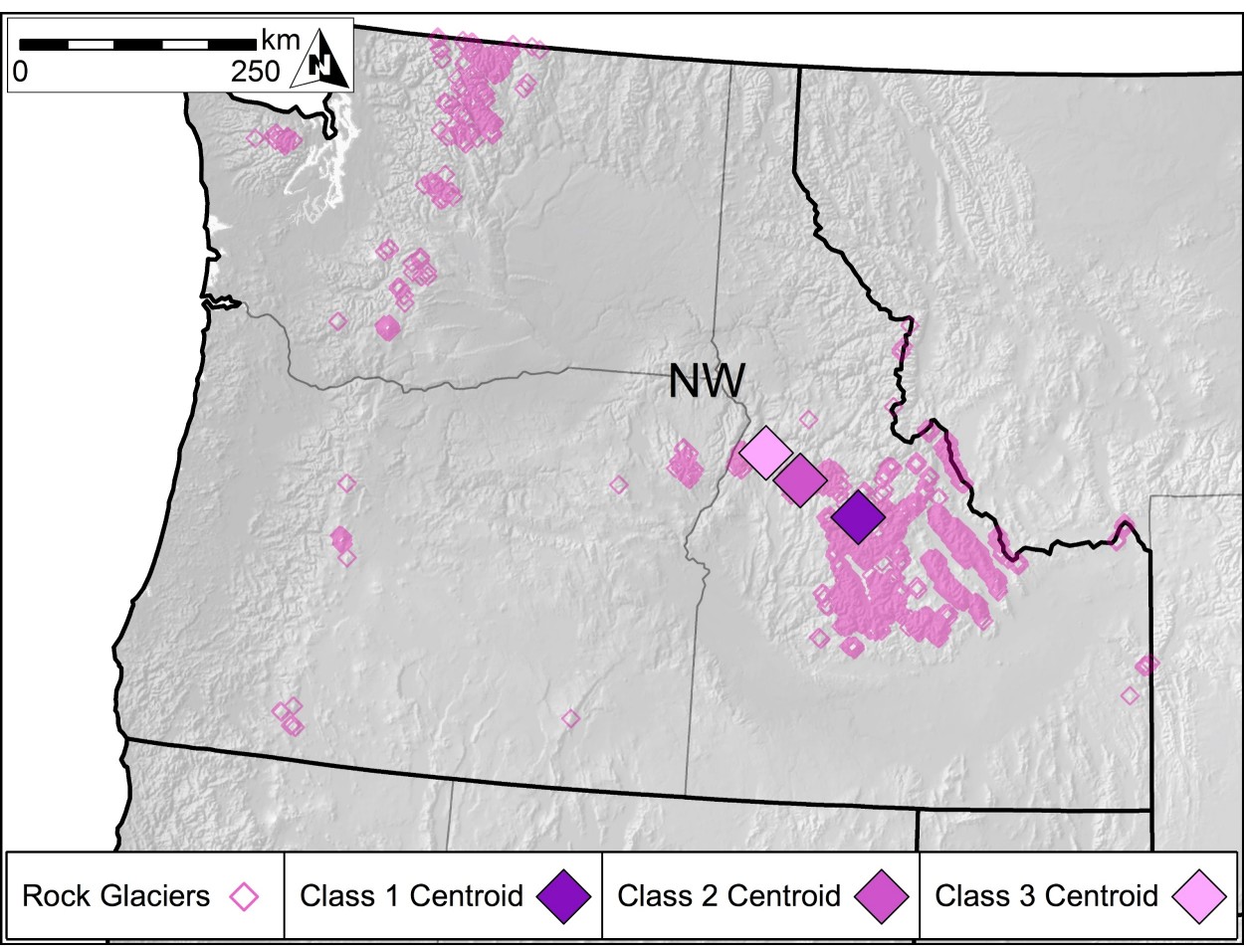

**Figure 6: Locations of NW Region rock glacier inventory features (n = 1993), as well as centroids for Class 1 (n = 1293), Class 2 (n = 512) and Class 3 (n = 188) features. Rock glaciers of the NW Region are largest and most densely concentrated in the Sawtooth Mountains of Idaho.**

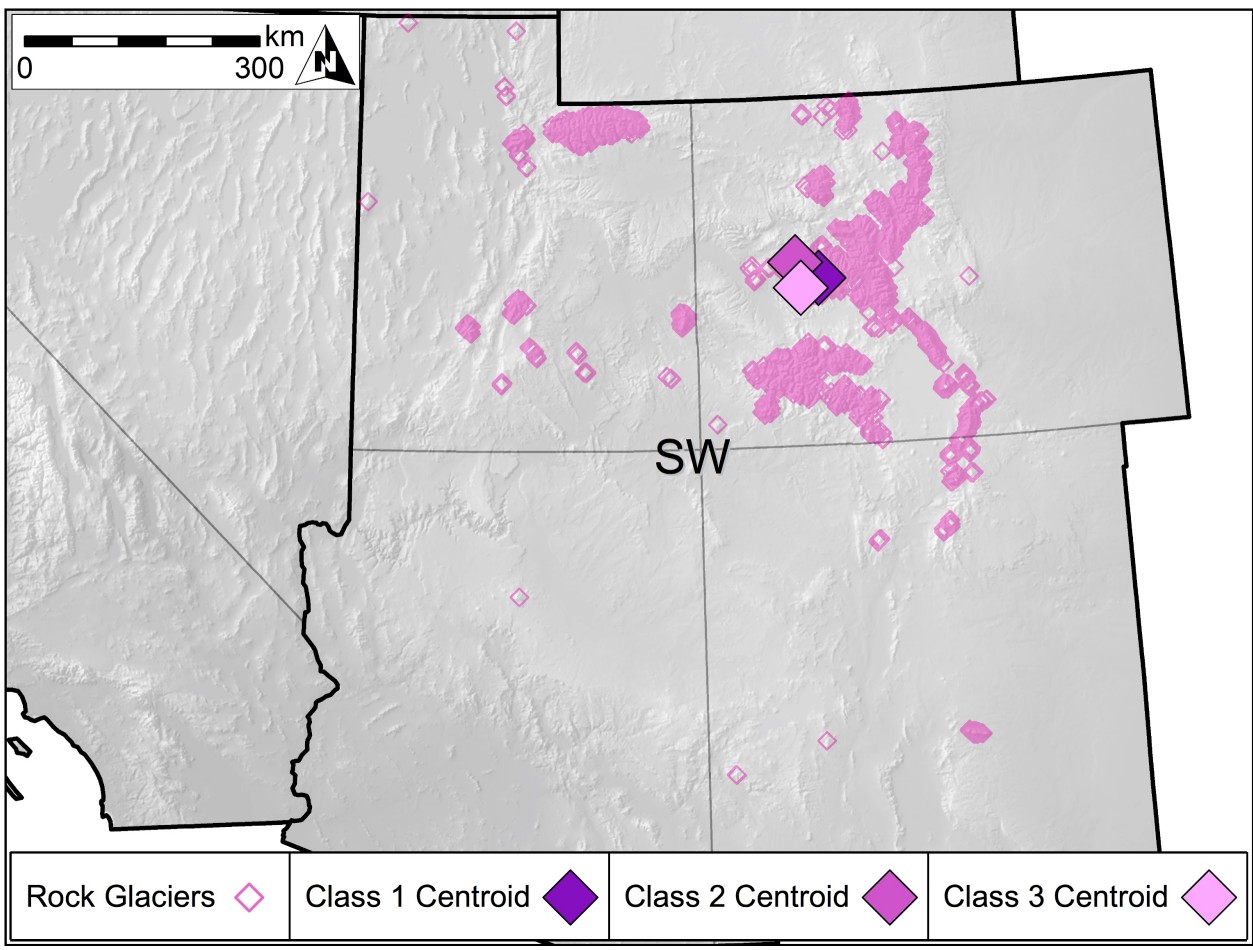


**Figure 7: Locations of SW Region rock glacier inventory features (n = 4870), as well as centroids for Class 1 (n = 3291), Class 2 (n = 1133) and Class 3 (n = 446) features. Rock glaciers of the SW Region are largest and most densely concentrated in the Front Range and San Juan Mountains of Colorado and the Uinta Mountains of Utah.**

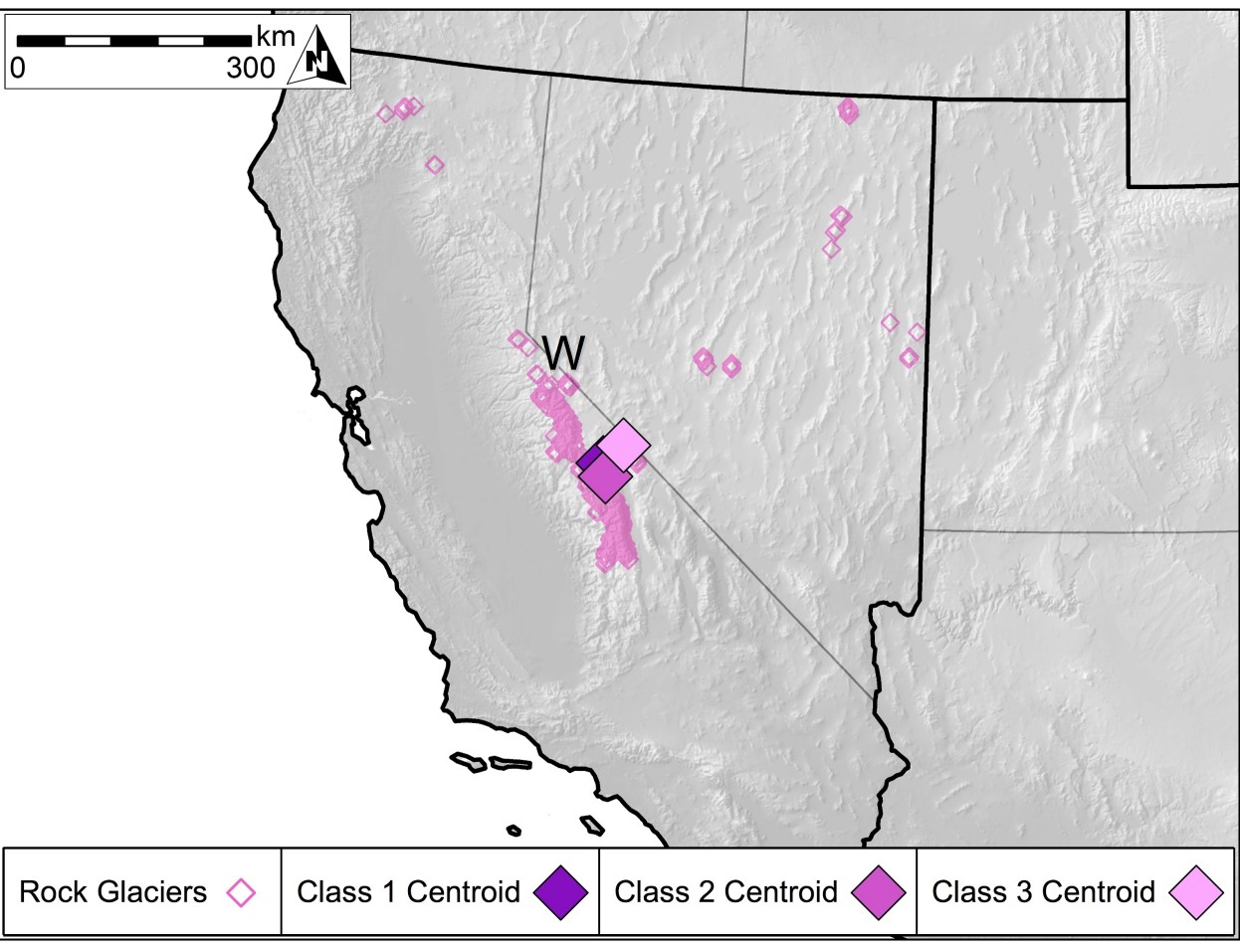

**Figure 8: Locations of W Region rock glacier inventory features (n = 817), as well as centroids for Class 1 (n = 552), Class 2 (n = 181) and Class 3 (n = 84) features. Rock glaciers of the W Region are largest and most densely concentrated in the Sierra Nevada of California.**


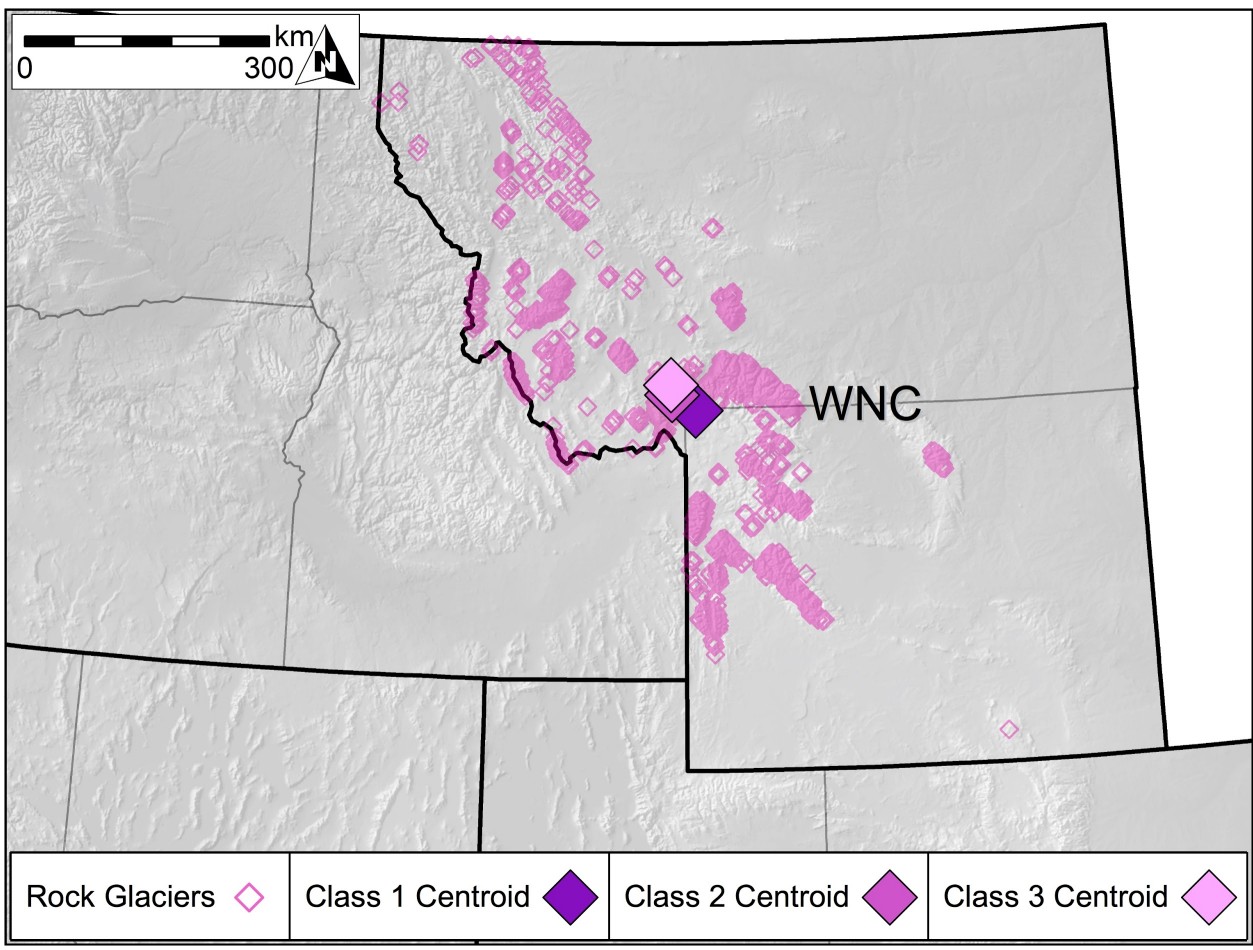

Rock Glaciers ◇ | Class 1 Centroid ◆ | Class 2 Centroid ◆ | Class 3 Centroid ◆

**Figure 9: Locations of WNC Region rock glacier inventory features (n = 2652), as well as centroids for Class 1 (n = 1906), Class 2 (n = 589) and Class 3 (n = 157) features. Rock glaciers of the WNC Region are largest and most densely concentrated in the Beartooth Mountains of Montana and the Absaroka Range of Wyoming.**


## Tables

**Table 1: Notable previous rock glacier inventories evaluated during comprehensive literature review. Only inventories that identified > 50 rock glaciers (i.e., at least regional scale) are included here, though sporadic smaller local inventories have been compiled.**

| Continent | Primary Investigator(s) | Region | Rock Glaciers Identified |
|---|---|---|---|
| Asia | Bolch and Gorbunov (2014) | Northen Tian Shan | 72 |
| Europe | Cremonese et al. (2011) | European Alps | 4795 |
| | Baroni et al. (2004) | Italian Alps | 216 |
| | Delaloye et al. (1998) | Swiss Alps | 321 |
| | Frauenfelder et al. (2005) | European Alps | 84 |
| | Imhof (1996) | Swiss Alps | 80 |
| | Kenner and Magnusson (2017) | Swiss Alps | 239 |
| | Lambiel and Reynard (2001) | Swiss Alps | 239 |
| | Magori et al. (2020) | Balkan Peninsula | 224 |
| | Scotti et al. (2013) | Italian Alps | 1514 |
| | Seppi et al. (2012) | Italian Alps | 705 |
| | Wagner et a. (2020a) | Austrian Alps | 5769 |
| North America | Millar and Westfall (2008) | Sierra Nevada | 289 |
| | Humlum (2000) | West Greenland | 400 |
| | Janke (2007) | U.S. Rocky Mountains | 220 |
| | Janke and Frauenfelder (2008) | U.S. Rocky Mountains | 180 |
| | Liu et al. (2013) | Sierra Nevada | 67 |
| South America | Angillieri (2010) | Argentine Andes | 155 |
| | Falaschi et al. (2014) | Argentine Andes | 488 |
| | Falaschi et al. (2015) | Patagonian Andes | 177 |
| | Rangecroft et al. (2014) | Bolivian Andes | 94 |

**Table 2: Rock glacier counts by NOAA Climate Region. The SW and WNC Regions account for nearly 73% of rock glaciers identified.**

| NOAA Region | Class 1 (count (mean area)) | Class 2 (count (mean area)) | Class 3 (count (mean area)) | Total Rock Glaciers (count (mean area)) |
|---|---|---|---|---|
| NW Region | 1293 (0.09 km$^2$) | 512 (0.05 km$^2$) | 188 (0.04 km$^2$) | 1993 (0.07 km$^2$) |
| SW Region | 3291 (0.12 km$^2$) | 1133 (0.05 km$^2$) | 446 (0.04 km$^2$) | 4870 (0.09 km$^2$) |
| W Region | 552 (0.16 km$^2$) | 181 (0.06 km$^2$) | 84 (0.05 km$^2$) | 817 (0.12 km$^2$) |
| WNC Region | 1906 (0.13 km$^2$) | 589 (0.06 km$^2$) | 157 (0.05 km$^2$) | 2652 (0.11 km$^2$) |
| All Regions | 7042 (0.12 km$^2$) | 2415 (0.05 km$^2$) | 875 (0.04 km$^2$) | 10,332 (0.10 km$^2$) |

**Table 3: Moran's I statistics for rock glacier class. Spatial clustering is most severe in the W Region.**

| NOAA Region | Moran's Index | z-score | p-value | Pattern |
|---|---|---|---|---|
| NW Region | 0.100 | 3.904 | < 0.001 | Clustered |
| SW Region | 0.099 | 8.596 | < 0.001 | Clustered |
| W Region | 0.176 | 4.179 | < 0.001 | Clustered |
| WNC Region | 0.119 | 5.982 | < 0.001 | Clustered |
| All Regions | 0.106 | 11.686 | < 0.001 | Clustered |

**Table 4: Moran's I statistics for rock glacier area. Spatial clustering is most severe in the W Region.**

| NOAA Region | Moran's Index | z-score | p-value | Pattern |
|---|---|---|---|---|
| NW Region | 0.159 | 6.228 | < 0.001 | Clustered |
| SW Region | 0.101 | 8.902 | < 0.001 | Clustered |
| W Region | 0.175 | 4.184 | < 0.001 | Clustered |
| WNC Region | 0.116 | 6.095 | < 0.001 | Clustered |
| All Regions | 0.116 | 6.905 | < 0.001 | Clustered |

**Table 5: Portland State University Active Rock Glacier Inventory shapefile attribute data dictionary.**

| Attribute Name | Attribute Description | Attribute Units |
|---|---|---|
| RG_CLASS | Rock Glacier Class | Class 1, 2, or 3 |
| AREA_KM2 | Rock Glacier Area | Square Kilometers |
| LAT | Centroid Latitude | WGS84 Decimal Degrees |
| LONG | Centroid Longitude | WGS84 Decimal Degrees |
| STATE | Centroid U.S. State | U.S. State Abbreviation |
| NOAA | NOAA Climate Region | NW, SW, W, or WNC |
| ELEV | Mean Elevation | Meters |
| SLOPE | Mean Slope | Degrees |
| EAST | Aspect Eastness | Unitless |
| NORTH | Aspect Northness | Unitless |
| RAD_WIN | Average Winter (December, January, February) Solar Radiation | Watt-hours Per Square Meter |
| RAD_SPR | Average Spring (March, April, May) Solar Radiation | Watt-hours Per Square Meter |
| RAD_SUM | Average Summer (June, July, August) Solar Radiation | Watt-hours Per Square Meter |
| RAD_FAL | Average Fall (September, October, November) Solar Radiation | Watt-hours Per Square Meter |
| RAD_ANN | Average Annual Solar Radiation | Watt-hours Per Square Meter |
| PPT_WIN | Average Winter (December, January, February) Precipitation | Millimeters |
| PPT_SPR | Average Spring (March, April, May) Precipitation | Millimeters |
| PPT_SUM | Average Summer (June, July, August) Precipitation | Millimeters |
| PPT_FAL | Average Fall (September, October, November) Precipitation | Millimeters |
| PPT_ANN | Average Annual Precipitation | Millimeters |

| SNO_WIN | Average Winter (December, January, February) Snowfall | Millimeters Water Equivalent |
|---|---|---|
| SNO_SPR | Average Spring (March, April, May) Snowfall | Millimeters Water Equivalent |
| SNO_SUM | Average Summer (June, July, August) Snowfall | Millimeters Water Equivalent |
| SNO_FAL | Average Fall (September, October, November) Snowfall | Millimeters Water Equivalent |
| SNO_ANN | Average Annual Snowfall | Millimeters Water Equivalent |
| TDMEAN_WIN | Average Winter (December, January, February) Dewpoint Temperature | Degrees Celsius |
| TDMEAN_SPR | Average Spring (March, April, May) Dewpoint Temperature | Degrees Celsius |
| TDMEAN_SUM | Average Summer (June, July, August) Dewpoint Temperature | Degrees Celsius |
| TDMEAN_FAL | Average Fall (September, October, November) Dewpoint Temperature | Degrees Celsius |
| TDMEAN_ANN | Average Annual Dewpoint Temperature | Degrees Celsius |
| TMAX_WIN | Average Winter (December, January, February) Maximum Temperature | Degrees Celsius |
| TMAX_SPR | Average Spring (March, April, May) Maximum Temperature | Degrees Celsius |
| TMAX_SUM | Average Summer (June, July, August) Maximum Temperature | Degrees Celsius |
| TMAX_FAL | Average Fall (September, October, November) Maximum Temperature | Degrees Celsius |
| TMAX_ANN | Average Annual Maximum Temperature | Degrees Celsius |
| TMEAN_WIN | Average Winter (December, January, February) Mean Temperature | Degrees Celsius |
| TMEAN_SPR | Average Spring (March, April, May) Mean Temperature | Degrees Celsius |
| TMEAN_SUM | Average Summer (June, July, August) Mean Temperature | Degrees Celsius |
| TMEAN_FAL | Average Fall (September, October, November) Mean Temperature | Degrees Celsius |
| TMEAN_ANN | Average Annual Mean Temperature | Degrees Celsius |
| TMIN_WIN | Average Winter (December, January, February) Minimum Temperature | Degrees Celsius |
| TMIN_SPR | Average Spring (March, April, May) Minimum Temperature | Degrees Celsius |

| | | |
|---|---|---|
| TMIN_SUM | Average Summer (June, July, August) Minimum Temperature | Degrees Celsius |
| TMIN_FAL | Average Fall (September, October, November) Minimum Temperature | Degrees Celsius |
| TMIN_ANN | Average Annual Minimum Temperature | Degrees Celsius |
| VPDMAX_WIN | Average Winter (December, January, February) Maximum Vapor Pressure Deficit | Hectopascals |
| VPDMAX_SPR | Average Spring (March, April, May) Maximum Vapor Pressure Deficit | Hectopascals |
| VPDMAX_SUM | Average Summer (June, July, August) Maximum Vapor Pressure Deficit | Hectopascals |
| VPDMAX_FAL | Average Fall (September, October, November) Maximum Vapor Pressure Deficit | Hectopascals |
| VPDMAX_ANN | Average Annual Maximum Vapor Pressure Deficit | Hectopascals |
| VPDMEAN_WI | Average Winter (December, January, February) Mean Vapor Pressure Deficit | Hectopascals |
| VPDMEAN_SP | Average Spring (March, April, May) Mean Vapor Pressure Deficit | Hectopascals |
| VPDMEAN_SU | Average Summer (June, July, August) Mean Vapor Pressure Deficit | Hectopascals |
| VPDMEAN_FA | Average Fall (September, October, November) Mean Vapor Pressure Deficit | Hectopascals |
| VPDMEAN_AN | Average Annual Mean Vapor Pressure Deficit | Hectopascals |
| VPDMIN_WIN | Average Winter (December, January, February) Minimum Vapor Pressure Deficit | Hectopascals |
| VPDMIN_SPR | Average Spring (March, April, May) Minimum Vapor Pressure Deficit | Hectopascals |
| VPDMIN_SUM | Average Summer (June, July, August) Minimum Vapor Pressure Deficit | Hectopascals |
| VPDMIN_FAL | Average Fall (September, October, November) Minimum Vapor Pressure Deficit | Hectopascals |
| VPDMIN_ANN | Average Annual Minimum Vapor Pressure Deficit | Hectopascals |