# Peer review of "Active rock glaciers of the contiguous United States: GIS inventory and spatial distribution patterns"

_Earth System Science Data, 2020_

## Referee Comment (RC1) · Anonymous Referee #1 · 3 Nov 2020

General Comments: This contribution presents a nation-wide inventory of "intact" rock glaciers (sensu Barsch, 1996) and fully mantled debris-covered glaciers for the contiguous USA. The topic is suitable to ESSD. The authors justify their work with the need for continental-scale inventories, which are currently not available. This is clearly an impressive mapping effort. On the down side, I find the mapping rules adopted, the inherent mapping uncertainty and the metadata specifics to be insufficiently illustrated (i.e., with figures and photos) and documented. I also note a number of drawbacks in the inventorying approach that need to be considered carefully, before this database maybe considered for further analysis. In particular, the typology of landforms (i.e., intact rock glaciers and fully mantled debris-covered glaciers) blended in and the dy-

namic classification scheme adopted, makes the present inventory not comparable with other existing inventories around the world. For these reasons, making statistical inference from this database in its present form may lead to misleading conclusions.

Major points to be addressed are summarized under the following headings:

1. Rock glaciers and debris-covered glaciers: Although I agree that making a clearcut distinction between rock glaciers and debris-covered glaciers in some cases is subject to large uncertainties, which can only be resolved with direct geophysical investigation, a number of morphological features are known to be distinctive of debris-covered glaciers. These include, but are not limited to, the presence of crevasses with exposed ice, ice cliffs, abundant thermokarst and supraglacial lakes, supraglacial streams, outflow breaches. In this regard, adding some sample images showing which kind of debris-covered glaciers were excluded from the inventory would help the reader a lot. I suggest that the authors add a field in the PSURGI attribute table indicating whether a given polygon is a rock glacier, a debris-covered glacier, or uncertain i.e., when they are unable to distinguish between the two.

2. Degree of activity classification scheme: The dynamic classification scheme adopted in PSURGI subdivides intact rock glaciers into three classes: highly, intermediately, and minimally active. This approach makes PSURGI not immediately comparable with most of existing inventories around the world, which discriminate intact rock glaciers into inactive (i.e., no front movement) and active landforms (Barsch, 1996). Recent mapping tests in Northern Tyrol have shown that distinction between active and inactive rock glaciers is subject to high uncertainty, and that inactive rock glaciers (those that supposedly should move more slowly) displayed large disagreement among a pool of international, experienced mappers (Brardinoni et al., 2019). In this context, subdividing intact rock glaciers into three categories (as opposed to the classical two) appears unreliable. Along these lines, PSURGI approach to dynamic classification seems contradictory: on one hand it is stated that visual interpretation of imagery does not afford distinction between rock glaciers and debris-covered glaciers, on the other

hand, this same procedure would allow to discriminate three subtypes of intact rock glaciers. I believe that this type of fine distinction could be achieved reliably only with the aid of InSAR technology. For the reasons outlined above, I suggest that the authors revert their dynamic classification scheme for intact rock glaciers to the classical one (i.e., active and inactive).

3. Completeness: The question of inventory completeness is only brushed upon. A similar large-scale inventory should be coupled by a systematic testing on the variability and uncertainty among mappers involved in the inventory. For example, Google Earth Imagery, when not complemented by LiDAR-derived hillshades and high-resolution orthophoto mosaics has been shown to yield incomplete rock glacier detection, especially due to poor distinction between adjacent coalescing lobes (Brardinoni et al., 2019). In this context, the question of complex multi-lobe (or polymorphic) rock glaciers and the way in which these morphologies were mapped is not addressed. No example was provided. Any geomorphologist familiar with rock glacier mapping is aware of the inherent uncertainties associated with an inventory, yet the authors depict PSURGI as greatly accurate. Please consider tuning down some sentences in that section.

4. Rock glacier delineation (mapping rules): No specific description of the mapping rules applied in PSURGI is provided, and only vague wordy descriptions are given. For example, one of the most problematic issues when delineating a rock glacier polygon is typically represented by the extent of the rooting zone, which borders the upper end of a rock glacier. In the manuscript, I could not find which mapping rule has been applied to delineate the upper end of rock glaciers and exclude the rooting zone (assuming this was excluded from the mapped polygons). In Figure 1, class 2 example, the upper end of the polygon cuts across flow lines, following no apparent discontinuity in curvature or roughness. Was the mapping confidence consistent across the entire perimeter of this polygon? Overall, the three examples provided in Figure 1 do not struck me for being indicative of accurate mapping. Please add more examples and/or refine the outlines of the current ones.

5. Metedata: A database submitted for publication should come with well-documented metadata, including; i) A list of attributes in table format (i.e., the attribute table in the shapefile includes the dynamic classification only). ii) a list of complementary imagery used other than Google Earth Pro (i.e., currently the authors state the following in lines 83-85: "... supplementing with other plan-view imagery imported into ArcMap 10.4 when Google Earth Pro imagery was unsuitable due to cloud cover or other issues."

Specific Comments:

Title: considering the nature of the inventory, the title of the paper should acknowledge the inclusion of (intact) rock glaciers and fully mantled debris-covered glaciers.

Lines 23-25: "Two lesser known components of the montane cryosphere are rock glaciers and debris-covered glaciers, though presently there are no widely accepted formal definitions of either feature type that can be used to universally and unambiguously discriminate the two for all purposes"

In my opinion, this approach leads to confusion. The statement is not supported by any reference and discounts decades of research focused respectively on debris-covered glaciers and rock glaciers. It is more fair to say that there are widely accepted definitions of rock glaciers and debris-covered glaciers, and that a minority disagrees.

Lines 37-38: "Fully debris-covered glaciers are indistinguishable from the more traditionally defined rock glaciers through surface analysis alone".

Please try to support similar clearcut statements with references and illustrative examples (i.e., figures).

Lines 41-42: "The semantics of classifying these two cryospheric feature types is occasionally debated, but is not something we seek to resolve with this inventory (Clark et al. 1998, Potter 1972, Haeberli et al. 2006, Berthling 2011)."

I think the question is way beyond semantics. I understand that you do not want to enter into this dispute, but there are a number of morphological attributes that in many
instances should aid guiding distinction between rock glaciers and debris-covered glaciers. Merging rock glaciers and debris-covered glaciers in one database without any sort of morphological distinction represents a major limitation of this inventory.

Lines 131-134: "To partially address this ambiguity all features identified as rock glaciers were subsequently assigned to a three-tier classification system based on surface characteristics known to correlate with downslope movement motivated by deformation of the internal ice-rock matrix (Figure 1)."

Which would be the surface characteristics known to correlate with downslope movement? Please provide empirical data or reference to empirical publications showing such correlations. My impression is that by increasing the number of activity classes (three in this case), one is going to increase the degree of uncertainty. Please see my general comment #2.

Line 155: "after removing 146 small (< 0.01 km2) Class 3 rock glaciers following glaciological convention of area thresholds".

Why using this threshold size? Rock glaciers are not glaciers, neither are included in the World Glacier Inventory.

Conclusions (lines 252-260): most of the conclusions paragraph reads more like introduction. Please consider rewriting and connecting the conclusions to the main results outlined in the manuscript.

List of existing inventories:

I suggest adding the following references to the list of existing inventories, the first includes 5769 rock glaciers across Austria:

Wagner et al 2020. The first consistent inventory of rock glaciers and their hydrological catchments of the austrian alps. Austrian Journal of Earth Sciences, 113: 1-23.

Brigitte Magori, Petru Urdea, Alexandru Onaca & Florina Ardelean (2020) Distribution

and characteristics of rock glaciers in the Balkan Peninsula, Geografiska Annaler: Series A, Physical Geography, DOI: 10.1080/04353676.2020.1809905

---

## Referee Comment (RC2) · Anonymous Referee #2 · 4 Nov 2020

"general comments"

The contribution about intact rock glaciers (and fully mantled debris-covered glaciers) presented for (most of?) the contiguous USA is suitable for ESSD, very interesting and timely, as such inventories are much needed. However, a number of limitations of the presented inventory need to be stated clearly and also justified. The current inventory cannot be called a "rock glacier inventory", because it neglects relict rock glaciers but includes debris-covered glaciers. Moreover a classification scheme introduced herein makes a comparison to previous inventories difficult.

"specific comments"

[Figure]

lines 86 ff.: "quickly found no evidence of rock glaciers east of the Rocky Mountain States, therefore we focused our efforts on the 11 westernmost states (AZ, CA, CO, ID, MT, NM, NV, OR, UT, WA, WY)."

Besides using abbreviations for the US states without defining them, this is a statement that cannot be supported easily and needs further discussion. A search for rock glaciers in the Appalachian Mountains will allow you to find literature about potential "sightings": e.g. Putnam & Putnam (2009) reporting about inactive (!) and relict rock glaciers in northern Maine. Please justify why these landforms are excluded in the current inventory.

Lines 95 ff: "Because glaciers and rock glaciers are often co-located. . ."

Is this really always a true statement? In mountainous regions, a glacier will generally form if temperature, etc. allow, but also only if abundant precipitation is available; rock glaciers on the other hand will ask for relatively dryer regions. This aspect should be mentioned and potentially starting in locations where only glaciers are, might not be justified.

Lines 111 ff: "We focused our inventory efforts on identifying rock glaciers that, superficially, appear to contain appreciable internal ice fractions and are presently or were recently flowing downslope. . . . a second major distinction between our rock glacier inventory and classification system and other previous U.S. rock glacier inventory efforts is that we intentionally attempt to exclude relict rock glaciers."

Reading the manuscripts title as well as previous lines, the exclusion of relict rock glaciers is not really expected up to this point. The authors cannot provide a "rock glacier inventory" without relict rock glaciers. There would have to be a different title at least ("intact rock glacier inventory"?). However, I feel there is a general flaw in the approach, as was mentioned by the first referee's comments: on the one hand, a distinction in 3 activity classes is made, but relict rock glaciers are excluded and on the other hand potentially debris-covered glaciers are included without much of a

discussion about uncertainty related to this chosen approach. What kind of inventory is it then and how can it be compared to previous inventories in the USA and other inventories around the world (see the two mentioned inventories of referee #1!)? At least this needs to be mentioned first and then justified somehow (although I have a hard time to come up with a good explanation myself).

Please note also that actually the distribution of relict rock glaciers is especially of interest, as it is a great opportunity to understand climate and paleoclimate evolution. Moreover, from a hydrogeological viewpoint, intact as well as relict rock glaciers are of great interest and neglecting some of them (the relict ones) would make the current inventory only partially useful.

Please refer to e.g. Hayashi et al. (2019: " Alpine hydrogeology: The critical role of groundwater in sourcing the headwaters of the world") and Wagner et al. (2020: "Active rock glaciers as shallow groundwater reservoirs, Austrian Alps") besides Jones et al. (2019b) to appreciate the value of rock glaciers in general (with or without ice being present) for hydrologists, hydrogeologists, ecologists, water resource managers, etc.

Lines 133 ff: "Understandably, there can be some disagreement between analysts regarding rock glacier classification."

Besides the actual classification (about the issue of 3 instead of the usual 2 classes, please refer to reviewer one) shouldn't there be a word or two about the actual issue of delineation of rock glaciers (see e.g. Brardinoni et al., 2019 or Schmid et al., 2015). Moreover, the 3 classes seem to favor active rock glaciers and by neglecting relict rock glaciers, I suppose that quite a number of inactive rock glaciers are "lost" using the approach described herein. E.g. refer to Colucci et al (2019): " Is that a relict rock glacier?".

When considering all the above mentioned limitations with this inventory and by fully agreeing to all the very constructive criticism of the anonymous referee #1 (from the 3rd of November 2020), the actual results of the inventory seem somewhat "biased" to

say the least.

The actual results presented herein are moreover hard to judge, because the available shape file (PSURGI; https://doi.pangaea.de/10.1594/PANGAEA.918585) has no attributes attached to, besides the activity classification scheme (1 to 3). Is it planned to add the related attributes to this data set? Also with this dataset, the title does not include the information that no relict rock glaciers are included and will misguide the potential user of this dataset.

Allow me to jump directly to the Conclusion [as the discussion section about Inventory Accuracy seems to be guided by much confidence and might need some more cautious rewording (refer here to the comments of referee #1)]:

Line 252 ff: "We present the most spatially extensive geospatial rock glacier inventory in the world to date, a powerful tool informing a wide range of research and management applications."

Is this really true? Is this really a complete rock glacier inventory? IMHO there needs to be a clear differentiation between what has been done here and what previous rock glacier inventories tried to achieve. The current state of the PSURGI inventory does not allow a direct comparison to previous inventories, due to the different classification as well as the disregard of relict rock glaciers.

"technical corrections"

Lines 88 ff: "the 11 westernmost states (AZ, CA, CO, ID, MT, NM, NV, OR, UT, WA, WY)" Abbreviations should be explained first time they are used. Not everyone might be familiar with the US states abbreviations.

Figure 1: Class 1, 2 and 3 examples are not ideal, simply because the scale of the examples used is very different (factor 5).

Figure 2: Why not include the Sates boundaries so that the reader less familiar with them can relate to them? Color-coding of the mean (?) elevation of each landform

would allow the reader to appreciate the intact rock glacier distribution.

Figures 3 & 4: box-whisker plots presented here: are the whiskers the 10/90 percentiles or 5/95? Please add this information so that the outliers can be related accordingly.

Figures 5-9: Intact rock glacier density maps would paint a better picture than plotting centroids. Please reconsider the actual value of the current figures 5-9.

Table 1: The inventories by Kellerer-Pirklbauer et al (2012) and Krainer and Ribis (2012) are in the meantime replaced by a consistent rock glacier inventory of Austria (Wagner et al., 2020); see notes of referee #1 about this inventory and the one available for the Balkan Peninsula by Magori et al (2020). Moreover, the table would be of greater value if regions would be mentioned; e.g. Seppi et al. (2012): Eastern Italian Alps (Trentino).

―――――――――――――――――――

---

## Author Comment (AC1) · 1 Feb 2021

Referee comments in [[DOUBLE BRACKETS]], author responses in {{DOUBLE BRACES}}.

[[Anonymous Referee #1

General Comments: This contribution presents a nation-wide inventory of "intact" rock glaciers (sensu Barsch, 1996) and fully mantled debris-covered glaciers for the contiguous USA. The topic is suitable to ESSD. The authors justify their work with the need for continental-scale inventories, which are currently not available. This is clearly

an impressive mapping effort. On the down side, I find the mapping rules adopted, the inherent mapping uncertainty and the metadata specifics to be insufficiently illustrated (i.e., with figures and photos) and documented. I also note a number of drawbacks in the inventorying approach that need to be considered carefully, before this database maybe considered for further analysis. In particular, the typology of landforms (i.e., intact rock glaciers and fully mantled debris-covered glaciers) blended in and the dynamic classification scheme adopted, makes the present inventory not comparable with other existing inventories around the world. For these reasons, making statistical inference from this database in its present form may lead to misleading conclusions.

Major points to be addressed are summarized under the following headings:

1. Rock glaciers and debris-covered glaciers: Although I agree that making a clearcut distinction between rock glaciers and debris-covered glaciers in some cases is subject to large uncertainties, which can only be resolved with direct geophysical investigation, a number of morphological features are known to be distinctive of debris-covered glaciers. These include, but are not limited to, the presence of crevasses with exposed ice, ice cliffs, abundant thermokarst and supraglacial lakes, supraglacial streams, outflow breaches. In this regard, adding some sample images showing which kind of debris-covered glaciers were excluded from the inventory would help the reader a lot. I suggest that the authors add a field in the PSURGI attribute table indicating whether a given polygon is a rock glacier, a debris-covered glacier, or uncertain i.e., when they are unable to distinguish between the two.]]

{{Thank you for highlighting an ambiguity be believed was clear; extensive text has been added to the revised manuscript to reassure readers that very few, if any, debris-covered glaciers are likely to have been inadvertently included. All features were visually reviewed once more and 11 were deleted from the inventory as being likely debris-covered glaciers. Additionally, we have adopted a clear distinction throughout that the focus of this inventory is active rock glaciers. Of the features distinctive of debris-covered glaciers you list, none were observed on features that had not already

been excluded from the inventory for having expansive bare glacial ice in their accumulation zones. Original manuscript text clearly identifies "fully mantled debris-covered glaciers" as those which may inadvertently be included in the inventory, and clarifies that we followed previous studies and "omit features with expansive bare glacial ice in their accumulation zones as those are clearly debris-covered glaciers" (section 2.2). Additionally, and by definition, any debris-covered glacier that is "fully mantled" with regolith would not appear to have any crevasses with exposed ice or ice cliffs visible in all but the very highest resolution satellite imagery (i.e., sub-meter resolution) which is not widely and freely available, and was not used to create this inventory. As no discrimination between "fully mantled debris-covered glaciers" inadvertently included (i.e., fully mantled debris-covered glaciers that lack expansive surfaces of exposed ice in their accumulation zones or obvious supraglacial lakes/streams) can be confidently made though surface analysis alone given data limitations, no additional attribute data will be added. The operative term in describing how we addressed "ambiguous" debris covered glaciers is "fully mantled", and hopefully that is made clear in the revised text.}}

[[2. Degree of activity classification scheme: The dynamic classification scheme adopted in PSURGI subdivides intact rock glaciers into three classes: highly, intermediately, and minimally active. This approach makes PSURGI not immediately comparable with most of existing inventories around the world, which discriminate intact rock glaciers into inactive (i.e., no front movement) and active landforms (Barsch, 1996). Recent mapping tests in Northern Tyrol have shown that distinction between active and inactive rock glaciers is subject to high uncertainty, and that inactive rock glaciers (those that supposedly should move more slowly) displayed large disagreement among a pool of international, experienced mappers (Brardinoni et al., 2019). In this context, subdividing intact rock glaciers into three categories (as opposed to the classical two) appears unreliable. Along these lines, PSURGI approach to dynamic classification seems contradictory: on one hand it is stated that visual interpretation of imagery does not afford distinction between rock glaciers and debris-covered glaciers, on the other hand, this same procedure would allow to discriminate three subtypes of intact rock

glaciers. I believe that this type of fine distinction could be achieved reliably only with the aid of InSAR technology. For the reasons outlined above, I suggest that the authors revert their dynamic classification scheme for intact rock glaciers to the classical one (i.e., active and inactive).]]

{{We believe this inventory can indeed be readily and directly compared with any other inventories that identify "active" rock glaciers, provided those inventories also took the care to either provide an "active/inactive" attribute in any spatial data made available, or discriminate "active/inactive" features in their statistical analysis. We believe all features included in the inventory to fit accurately be described as "active" (i.e., exhibiting at least some flow annually), and are unaware of a universally accepted non-zero movement threshold that discriminates "active" form "inactive" rock glaciers. Numerous widely-cited rock glacier inventories use a variety of classification systems containing anywhere from six classes to a single class; two classes is by no means a global standard, and among those that do use two classes the specific class definitions and classification rules occasionally vary to a degree precluding direct comparison from one inventory to another. We agree InSAR or equivalent technology would be necessary if our classification system was quantitative and attempted to estimate actual flow rates, but the three-class system we employ is purely qualitative, based primarily on prevalence of ridge and swale surface banding and oversteepened terminal and lateral slopes. Ridge and swale surface banding is widely accepted to be the result of differential flow rates (i.e., absent from rock glaciers that are not presently or recently flowing), while oversteepened terminal and lateral slopes are widely accepted to be the result of cementing by flow motivated by the deformation of interstitial ice. We agree that there is much uncertainty and disagreement with respect to classifying rock glaciers, and state as much numerous times throughout the paper, but believe that our three-class system offers valuable flexibility to rock glacier researchers. Those more interested in active rock glaciers exhibiting the highest degree of classification certainty can focus on Class 1 features, while those more interested in probably currently active, though possibly recently inactive, rock glaciers and willing to accept a slightly lower degree of

classification certainty can focus on Class 3 features.}}

[[3. Completeness: The question of inventory completeness is only brushed upon. A similar large-scale inventory should be coupled by a systematic testing on the variability and uncertainty among mappers involved in the inventory. For example, Google Earth Imagery, when not complemented by LiDAR-derived hillshades and high-resolution orthophoto mosaics has been shown to yield incomplete rock glacier detection, especially due to poor distinction between adjacent coalescing lobes (Brardinoni et al., 2019). In this context, the question of complex multi-lobe (or polymorphic) rock glaciers and the way in which these morphologies were mapped is not addressed. No example was provided. Any geomorphologist familiar with rock glacier mapping is aware of the inherent uncertainties associated with an inventory, yet the authors depict PSURGI as greatly accurate. Please consider tuning down some sentences in that section.]]

{{As in the paper you reference, the systemic testing of classification variability we performed (Section 4.2) showed slightly less agreement among analysts from most active to least active features, but just as in the paper you reference, the systemic testing of classification variability we performed showed broad agreement for all three feature classes. Additionally, our inventory was compared to three other smaller regional inventories and showed high levels of agreement with each of them. We agree LiDAR-derived DEMs would likely increase the accuracy of this inventory, but LiDAR is not available for the vast majority of the study area. Text has been added to the manuscript to describe how we addressed multi-lobate rock glaciers (i.e., we link distinct accumulation zones to distinct rock glaciers), though it is worth noting that there is little agreement among rock glacier researchers on which features to focus on when classifying multi-lobate rock glaciers, or exactly when two adjacent lobes with a common accumulation zone should be considered two distinct rock glaciers. We are uncertain exactly which section you are referring to in your request that we "tune down some sentences in that section", but both the comparisons to other regional inventories and the comparisons to classifications between analysts support the conclusion that our

active rock glacier inventory is indeed quite accurate. Regardless, additional text has been added to revised manuscript to further clarify uncertainties and limitations.}}

[[4. Rock glacier delineation (mapping rules): No specific description of the mapping rules applied in PSURGI is provided, and only vague wordy descriptions are given. For example, one of the most problematic issues when delineating a rock glacier polygon is typically represented by the extent of the rooting zone, which borders the upper end of a rock glacier. In the manuscript, I could not find which mapping rule has been applied to delineate the upper end of rock glaciers and exclude the rooting zone (assuming this was excluded from the mapped polygons). In Figure 1, class 2 example, the upper end of the polygon cuts across flow lines, following no apparent discontinuity in curvature or roughness. Was the mapping confidence consistent across the entire perimeter of this polygon? Overall, the three examples provided in Figure 1 do not struck me for being indicative of accurate mapping. Please add more examples and/or refine the outlines of the current ones.]]

{{Since the three-class system we employ is purely qualitative, the mapping rules are also qualitatively described (Section 2.2). Despite your characterization of these mapping rules as "vague", these were the same rules described verbatim to the five independent analysts who systemically, and in isolation, tested our classification rules, which resulted in broad agreement between analysts for all three classes. With respect to rooting zone extents, we focused on sharp changes in slope, "from the steep slopes of exposed bedrock and unconsolidated talus in the rock glacier accumulation zone to the more gentle slope of the main body of the ice-thickened rock glacier". Text has been added to the manuscript to describe identification of upper rock glacier boundaries where exposed bedrock was not present as less confidently delineated than sharp lateral and terminal boundaries. The Class 2 example boundary in original Figure 1 (updated to Figure 2) does not cross any flow lines. Copious flow lines, which generally trend perpendicular to the fall line, are visible at the lower end of the rock glacier, but none are visible at the upper end. These three examples are broadly

representative of the three inventory class delineations overall and will not be edited or replaced.}}

[[5. Metedata: A database submitted for publication should come with well-documented metadata, including; i) A list of attributes in table format (i.e., the attribute table in the shapefile includes the dynamic classification only). ii) a list of complementary imagery used other than Google Earth Pro (i.e., currently the authors state the following in lines 83-85: "... supplementing with other plan-view imagery imported into ArcMap 10.4 when Google Earth Pro imagery was unsuitable due to cloud cover or other issues.")]]

{{Shapefile attributes have been added, and a descriptive attribute table has been added to the revised manuscript. Citation of additional plan-view imagery used has been added to the revised manuscript.}}

[[Specific Comments:

Title: considering the nature of the inventory, the title of the paper should acknowledge the inclusion of (intact) rock glaciers and fully mantled debris-covered glaciers.]]

{{Manuscript title, and text throughout revised manuscript, has been updated to reflect our focus on active rock glaciers. Considering the original text explained omission of "features with expansive bare glacial ice in their accumulation zones as those are clearly debris-covered glaciers", and the lack of supraglacial lakes or streams observed on remaining features retained, we are confident that very few "fully mantled debris-covered glaciers" were included in the inventory. Regardless, extensive additional text added throughout the revised manuscript to make our attempts to exclude "fully mantled debris-covered glaciers" even more explicit. Referencing "debris-covered" glaciers in the title when the relevant caveats are discussed numerous times in the manuscript does little to assist the intended audience, and we are confident that very few debris-covered glaciers are likely to have been inadvertently included. Given no further discrimination between the "fully mantled debris-covered glaciers" that may have been inadvertently included (i.e., those with no visible crevasses, ice walls, exposed glacial

ice, or supraglacial lakes/streams) and active rock glaciers can be confidently made through surface analysis alone given data limitations, and no attribute data will be added.}}

[[Lines 23-25: "Two lesser known components of the montane cryosphere are rock glaciers and debris-covered glaciers, though presently there are no widely accepted formal definitions of either feature type that can be used to universally and unambiguously discriminate the two for all purposes"

In my opinion, this approach leads to confusion. The statement is not supported by any reference and discounts decades of research focused respectively on debris-covered glaciers and rock glaciers. It is more fair to say that there are widely accepted definitions of rock glaciers and debris-covered glaciers, and that a minority disagrees.]]

{{Extensive text has been added to revised manuscript to describe the subset of debris covered glaciers (i.e., those with no visible crevasses, ice walls, exposed glacial ice, or supraglacial lakes/streams) that may have inadvertently been included due to our inability to discriminate them from active rock glaciers based on aerial/satellite imagery alone. Text and citations have been added to the manuscript to briefly describe the widely accepted "continuum concept" of glaciers, rock glaciers and debris-covered glaciers, as well as the common transition of debris-covered glaciers to rock glaciers. We will have to agree to disagree about how widely accepted your preferred definitions of debris-covered glaciers and rock glaciers are since you do not provide them in your comments here, but we are unaware of any rock glacier researchers who reject the "continuum concept" as wholly without merit. We agree that distinctions occasionally can, and when possible should, be made between rock glaciers and "fully mantled debris-covered glaciers" (i.e., those with no visible crevasses, ice walls, exposed glacial ice, or supraglacial lakes/streams), but such distinctions can only be confidently made with detailed field surveys (e.g., coring, GPR, etc.), cannot be made using manual aerial/satellite image classification, will ultimately come down to semantics based on internal ice fraction and arrangement, and should not be a barrier to disseminating this

active rock glacier inventory to the rock glacier research community.}}

[[Lines 37-38: "Fully debris-covered glaciers are indistinguishable from the more traditionally defined rock glaciers through surface analysis alone".

Please try to support similar clearcut statements with references and illustrative examples (i.e., figures).]]

{{We have added considerable clarifying/qualifying text to the passage you referenced, but the original point remains relevant and unchanged: fully mantled debris-covered glaciers (i.e., those with no visible crevasses, ice walls, exposed glacial ice, or supraglacial lakes/streams) are, in most cases, very difficult, if not impossible, to confidently discriminate from active rock glaciers through surface analysis alone.}}

[[Lines 41-42: "The semantics of classifying these two cryospheric feature types is occasionally debated, but is not something we seek to resolve with this inventory (Clark et al. 1998, Potter 1972, Haeberli et al. 2006, Berthling 2011)."

I think the question is way beyond semantics. I understand that you do not want to enter into this dispute, but there are a number of morphological attributes that in many instances should aid guiding distinction between rock glaciers and debris-covered glaciers. Merging rock glaciers and debris-covered glaciers in one database without any sort of morphological distinction represents a major limitation of this inventory.]]

{{We appreciate you comment, but fear you have drastically overestimated the number of debris covered glaciers that have been inadvertently included in our active rock glacier inventory. See our numerous responses above referencing imagery limitations with respect to confidently discriminating all "fully mantled debris-covered glaciers" (i.e., those with no visible crevasses, ice walls, exposed glacial ice, or supraglacial lakes/streams) from active rock glaciers, as well as extensive text expansion on this topic in the introduction of the revised manuscript. Again, we are confident that very few debris covered glaciers were included in the active rock glacier inventory.}}

[[Lines 131-134: "To partially address this ambiguity all features identified as rock glaciers were subsequently assigned to a three-tier classification system based on surface characteristics known to correlate with downslope movement motivated by deformation of the internal ice-rock matrix (Figure 1)."

Which would be the surface characteristics known to correlate with downslope movement? Please provide empirical data or reference to empirical publications showing such correlations. My impression is that by increasing the number of activity classes (three in this case), one is going to increase the degree of uncertainty. Please see my general comment #2.]]

{{Ridge and swale surface banding, commonly referred to as "flow banding", is widely accepted to be the result of differential flow rates and is highlighted in virtually every peer reviewed publication that focuses on rock glacier flow rates. We believe this correlation is patently obvious to any rock glacier researchers, our intended audience, but have added several citations to Section 2.2 to address the reviewer's concern. More flow banding indicates more flow, which in turn indicates higher levels of rock glacier activity. We are confident that all features included in the inventory can accurately be described as "active", and by providing additional qualitative information in the form of our our three-tired classification scheme we have added flexibility to future applications of the inventory. As demonstrated in our blind tests of classifications completed by numerous analysts, classification homogeneity is high across all three classes, but highest for Class 1 features. If readers wish to further analyze the inventory and only include those features with the very highest classification confidence, they can focus on Class 1 features. If readers wish to further analyze the inventory and are satisfied with slightly lower classification confidence, they can include all three classes of features.}}

[[Line 155: "after removing 146 small (< 0.01 km2) Class 3 rock glaciers following glaciological convention of area thresholds".

Why using this threshold size? Rock glaciers are not glaciers, neither are included in

the World Glacier Inventory.]]

{{Rock glacier research is very much informed by glacier research, a discipline with much more robust inventories available due to much more robust remote sensing analysis techniques available, mostly stemming from the spectral reluctance of exposed ice. Glacier inventories have identified a lower area threshold beyond which glaciers cannot be confidently identified and delineated, and given rock glaciers are much more difficult to identify than glaciers it seems only prudent that we apply the same area threshold. In any event, the omitted features would have contributed only a minuscule fraction of the features included by count, and a virtually infinitesimal fraction of the features included by area.}}

[[Conclusions (lines 252-260): most of the conclusions paragraph reads more like introduction. Please consider rewriting and connecting the conclusions to the main results outlined in the manuscript.]]

{{Conclusions section in revised manuscript has been expanded and refocused as you suggest.}}

[[List of existing inventories:

I suggest adding the following references to the list of existing inventories, the first includes 5769 rock glaciers across Austria:

Wagner et al 2020. The first consistent inventory of rock glaciers and their hydrological catchments of the austrian alps. Austrian Journal of Earth Sciences, 113: 1-23.

Brigitte Magori, Petru Urdea, Alexandru Onaca & Florina Ardelean (2020) Distribution and characteristics of rock glaciers in the Balkan Peninsula, Geografiska Annaler: Series A, Physical Geography, DOI: 10.1080/04353676.2020.1809905]]

{{Thank you for the suggestion, both references have been added to the revised manuscript.}}

[[Anonymous Referee #2

"general comments"

The contribution about intact rock glaciers (and fully mantled debris-covered glaciers) presented for (most of?) the contiguous USA is suitable for ESSD, very interesting and timely, as such inventories are much needed. However, a number of limitations of the presented inventory need to be stated clearly and also justified. The current inventory cannot be called a "rock glacier inventory", because it neglects relict rock glaciers but includes debris-covered glaciers. Moreover a classification scheme introduced herein makes a comparison to previous inventories difficult.]]

{{As outlined in responses above, we have adopted the clear description of "active" rock glaciers to both the revised manuscript title and throughout the revised manuscript text. See also extensive responses above and extensive text added to the revised manuscript to assure readers that very few fully mantled debris-covered glaciers, and only those truly indistinguishable from active rock glaciers (i.e., those with no visible crevasses, ice walls, exposed glacial ice, or supraglacial lakes/streams) may have been inadvertently included. We appreciate your encouragement, but disagree that our inventory "cannot be called a rock glacier inventory" because it does not include inactive/relict rock glaciers. Nonetheless, as noted we have added the qualifying term "active rock glaciers" throughout the revised manuscript to eliminate any ambiguity. It is widely appreciated that identification of relict rock glaciers, especially by surface analysis, is incredibly difficult even when in the field, let alone through satellite image analysis, and there is often wide disagreement even among seasoned rock glacier researchers. We have developed the world's largest coherent active rock glacier inventory, both in study area and features identified, and made a conscious choice that is clearly identified in both the original and revised manuscripts to exclude relict rock glaciers. Forest, wetland, glacier, permafrost and virtually any other landform inventory does not necessarily need to include "relict" features (i.e., features that once were extant but no longer meet functional definitions) to be relevant or useful. As noted, we

have added "active" as a descriptor of the rock glaciers to the title and throughout the revised manuscript, but confidently identifying all relict rock glaciers of the contiguous United States through visual analysis of remote sensing imagery is at present a technical impossibility and should not be a barrier to disseminating this inventory to the rock glacier research community.}}

[["specific comments"

lines 86 ff.: "quickly found no evidence of rock glaciers east of the Rocky Mountain States, therefore we focused our efforts on the 11 westernmost states (AZ, CA, CO, ID, MT, NM, NV, OR, UT, WA, WY)."

Besides using abbreviations for the US states without defining them, this is a statement that cannot be supported easily and needs further discussion. A search for rock glaciers in the Appalachian Mountains will allow you to find literature about potential "sightings": e.g. Putnam & Putnam (2009) reporting about inactive (!) and relict rock glaciers in northern Maine. Please justify why these landforms are excluded in the current inventory.]]

{{Full state names have been added to the manuscript to eliminate any confusion. Rock glaciers of the Eastern U.S. were researched extensively during literature review, including the paper you reference, which only suggests rock glacier activity in the area over 10,000 years ago. The possible rock glaciers identified by Putnam & Putnam are heavily vegetated, have been inactive/relict for likely thousands of years, and (more pragmatically) would be impossible to identify from surface analysis of satellite imagery alone. Please view their "possible rock glacier sightings" in Google Earth or other aerial/satellite imagery for confirmation; these features could not have been identified without extensive fieldwork, and are not believed to be conclusive evidence for anything approaching an "active" rock glacier (the focus of this inventory) by us. To our knowledge, no research has suggested any rock glacier activity in the Eastern U.S. since the beginning of the Holocene. While we recognize there are a few relict rock

glaciers in the Eastern U.S., as clearly stated in the manuscript we focused on active rock glaciers and agree with "the genetic rock glacier definition, "the visible expression of cumulative deformation by long-term creep of ice/debris mixtures under permafrost conditions", proposed by Berthling (2011)." We cannot include features that present no visible evidence for deformation in aerial/satellite imagery in this inventory based entirely on visual analysis of aerial/satellite imagery.}}

[[Lines 95 ff: "Because glaciers and rock glaciers are often co-located. . ." Is this really always a true statement? In mountainous regions, a glacier will generally form if temperature, etc. allow, but also only if abundant precipitation is available; rock glaciers on the other hand will ask for relatively dryer regions. This aspect should be mentioned and potentially starting in locations where only glaciers are, might not be justified.]]

{{Yes, we believe the statement as written is true, that "glaciers and rock glaciers are often co-located", and nowhere assert that it is "always true", as in your comment. Rock glaciers are not always co-located with glaciers, but since they so often are, areas immediately surrounding glaciers and/or perennial snowpack were the best place to start our inventory. However, as stated throughout the manuscript and shown in figures, we quickly expanded our search areas far beyond the proximity of glaciers and/or perennial snowfields, and indeed most active rock glaciers identified were found dozens, if not hundreds, of kilometers from any extant glaciers and/or perennial snowfields. Starting our search for rock glaciers near glaciers and/or perennial snowfields in no way limited the final inventory results, but based on the overwhelming preponderance of evidence available in the relevant literature it was the most logical place to begin our search.}}

[[Lines 111 ff: "We focused our inventory efforts on identifying rock glaciers that, surficially, appear to contain appreciable internal ice fractions and are presently or were recently flowing downslope. . . . a second major distinction between our rock glacier inventory and classification system and other previous U.S. rock glacier inventory efforts is that we intentionally attempt to exclude relict rock glaciers."

Reading the manuscripts title as well as previous lines, the exclusion of relict rock glaciers is not really expected up to this point. The authors cannot provide a "rock glacier inventory" without relict rock glaciers. There would have to be a different title at least ("intact rock glacier inventory"?). However, I feel there is a general flaw in the approach, as was mentioned by the first referee's comments: on the one hand, a distinction in 3 activity classes is made, but relict rock glaciers are excluded and on the other hand potentially debris-covered glaciers are included without much of a discussion about uncertainty related to this chosen approach. What kind of inventory is it then and how can it be compared to previous inventories in the USA and other inventories around the world (see the two mentioned inventories of referee #1!)? At least this needs to be mentioned first and then justified somehow (although I have a hard time to come up with a good explanation myself).]]

{{Revised manuscript title and text have been extensively modified to include the relevant qualifier of "active" when describing the rock glaciers we intended to find and ultimately included in the inventory. As in our previous response, we disagree that inactive/relict features must be included for the active rock glacier inventory to be useful to the research community, but nonetheless have made extensive revisions and widespread use of the qualifier "active rock glacier" to remove any ambiguity. See above responses and extensive manuscript revisions for fuller explanation of numerous steps taken to exclude all debris-covered glaciers that could confidently be distinguished from active rock glaciers from the inventory. We believe this inventory can indeed be readily and directly compared with any other inventories that identify "active" rock glaciers, provided those inventories also took the care to either provide an "active/inactive" attribute in any spatial data made available, or discriminate "active/inactive" features in their statistical analysis. The omission of inactive/relict rock glaciers is well supported by previous research that shows how difficult, if not impossible, it is to confidently identify inactive/relict rock glaciers from visual analysis of aerial/satellite imagery. Extensive text further explaining this widely accepted reality has been added to the revised manuscript.}}

[[Please note also that actually the distribution of relict rock glaciers is especially of interest, as it is a great opportunity to understand climate and paleoclimate evolution. Moreover, from a hydrogeological viewpoint, intact as well as relict rock glaciers are of great interest and neglecting some of them (the relict ones) would make the current inventory only partially useful.]]

{{Please see numerous responses above. We appreciate the value of inventorying inactive/relict rock glaciers, but as noted numerous times in our responses as well as the original and revised manuscript texts, confidently identifying these features using the analysis techniques and data sets available to us is not feasible, a reality supported by the available literature. We recognize and appreciate the limitations of this active rock glacier inventory, but feel it is a tremendous step towards a full and complete understanding of rock glaciers of the contiguous U.S.}}

[[Please refer to e.g. Hayashi et al. (2019: " Alpine hydrogeology: The critical role of groundwater in sourcing the headwaters of the world") and Wagner et al. (2020: "Active rock glaciers as shallow groundwater reservoirs, Austrian Alps") besides Jones et al. (2019b) to appreciate the value of rock glaciers in general (with or without ice being present) for hydrologists, hydrogeologists, ecologists, water resource managers, etc.]]

{{While recognize relict rock glaciers are important for all the reasons you mention and several more, we respectfully disagree that we "cannot provide a "rock glacier inventory" without relict rock glaciers" for all the reasons outlined in our previous responses. The omission of relict rock glaciers was primarily a pragmatic one in that it is widely appreciated that identification of relict rock glaciers, especially by surface analysis alone is incredibly difficult even when in the field, let alone through aerial/satellite imagery analysis, and there is often wide disagreement even among seasoned rock glacier researchers.}}

[[Lines 133 ff: "Understandably, there can be some disagreement between analysts

regarding rock glacier classification."

Besides the actual classification (about the issue of 3 instead of the usual 2 classes, please refer to reviewer one) shouldn't there be a word or two about the actual issue of delineation of rock glaciers (see e.g. Brardinoni et al., 2019 or Schmid et al., 2015). Moreover, the 3 classes seem to favor active rock glaciers and by neglecting relict rock glaciers, I suppose that quite a number of inactive rock glaciers are "lost" using the approach described herein. E.g. refer to Colucci et al (2019): " Is that a relict rock glacier?".]]

{{Indeed, and as is explicitly stated throughout the original and revised manuscripts, inactive/relict rock glaciers were not "lost", but intentionally excluded. From the out-set, this inventory was designed to identify only active rock glaciers, as they are the only rock glaciers that can be confidently identified using the analysis techniques and data sets available to us. As also stated several times in the original and revised manuscripts, our hope is that this active rock glacier inventory will foster additional research, and is in no way intended to be the final word on rock glaciers of the contiguous U.S. despite its vast study area and large feature count.}}

[[When considering all the above mentioned limitations with this inventory and by fully agreeing to all the very constructive criticism of the anonymous referee #1 (from the 3rd of November 2020), the actual results of the inventory seem somewhat "biased" to say the least.]]

{{As in our responses above, as well as both the original and revised manuscripts, our intention was unambiguously stated as the identification of active rock glaciers. This intention was informed primarily by what was feasible, not what was ideal, yet we feel this inventory is a valuable contribution to the rock glacier research community despite the intentional omission of inactive/relict rock glaciers.}}

[[The actual results presented herein are moreover hard to judge, because the available shape file (PSURGI; https://doi.pangaea.de/10.1594/PANGAEA.918585) has no

attributes attached to, besides the activity classification scheme (1 to 3). Is it planned to add the related attributes to this data set? Also with this dataset, the title does not include the information that no relict rock glaciers are included and will misguide the potential user of this dataset.]]

{{Shapefile attributes have been added, and a descriptive attribute table has been added to the revised manuscript. Citation of additional plan-view imagery used has been added to the revised manuscript. We would be surprised if anyone attempted to use the rock glacier inventory geospatial data without also reading this manuscript, which makes clear throughout that our intention and inventory are both focused active rock glaciers, but additional descriptors have been added to the data download page.}}

[[Allow me to jump directly to the Conclusion [as the discussion section about Inventory Accuracy seems to be guided by much confidence and might need some more cautious rewording (refer here to the comments of referee #1)]:]]

{{The accuracy of our active rock glacier inventory is quantitatively estimated by both comparing our findings to previous smaller regional-scale rock glacier inventories and comparison between classifications by analysts who systemically, and in isolation, tested our classification rules. Both evaluations resulted in broad agreement between previous smaller regional-scale rock glacier inventories, as well as classifications individual analysts assigned based on our qualitative classification rules.}}

[[Line 252 ff: "We present the most spatially extensive geospatial rock glacier inventory in the world to date, a powerful tool informing a wide range of research and management applications."

Is this really true? Is this really a complete rock glacier inventory? IMHO there needs to be a clear differentiation between what has been done here and what previous rock glacier inventories tried to achieve. The current state of the PSURGI inventory does not allow a direct comparison to previous inventories, due to the different classification as well as the disregard of relict rock glaciers.]]

{{Nowhere in the original manuscript did we assert our rock glacier inventory was "complete", and went to considerable lengths to unambiguously state that we intentionally omitted inactive/relict rock glaciers due to the infeasibility of confidently identifying them using the analysis techniques and data sets available to us. Nonetheless, and as noted in our responses above, we have gone to even greater lengths in the revised manuscript to disabuse any potential reader of the notion that our active rock glacier inventory includes inactive/relict rock glaciers. As noted in our responses above, we believe this inventory can indeed be readily and directly compared with any other inventories that identify "active" rock glaciers, provided those inventories also took the care to either provide an "active/inactive" attribute in any spatial data made available, or discriminate "active/inactive" features in their statistical analysis.}}

[["technical corrections" Lines 88 ff: "the 11 westernmost states (AZ, CA, CO, ID, MT, NM, NV, OR, UT, WA, WY)" Abbreviations should be explained first time they are used. Not everyone might be familiar with the US states abbreviations.]]

{{Full state names have been added to the manuscript.}}

[[Figure 1: Class 1, 2 and 3 examples are not ideal, simply because the scale of the examples used is very different (factor 5).]]

{{The different spatial scales is actually one of the most critical points made by orginal Figure 1 (updated Figure 2), in that the most active rock glaciers are also generally much larger than the least active rock glaciers. The three examples shown are quite close to the average area for each class, a point also made in the upper left panel of original Figure 3 (updated Figure 4).}}

[[Figure 2: Why not include the Sates boundaries so that the reader less familiar with them can relate to them? Color-coding of the mean (?) elevation of each landform would allow the reader to appreciate the intact rock glacier distribution.]]

{{State boundaries are not shown in original Figure 2 (updated Figure 3) because the

focus of that figure is NOAA Climate Regions. State boundaries are shown original Figure 5 through Figure 8 (updated Figure 6 through Figure 9). Color coding over 10,000 points by elevation is meaningless when the features are highly clustered (often only tens of meters apart) and represented by symbols that span over ten kilometers across on the map. You will have to trust us, but we tried symbolizing the rock glacier icons by elevation and a dozen other variables long ago and the outputs are meaningless as you only see the one symbol drawn on "top" of all the other nearby symbols. See Fig. 4 of Wagner et al. 2020 for a good example of how little information the map symbology you suggest would provide.}}

[[Figures 3 & 4: box-whisker plots presented here: are the whiskers the 10/90 percentiles or 5/95? Please add this information so that the outliers can be related accordingly.]]

{{Text added to revised figure captions. Boxplot whiskers represent 1.5 times the interquartile range, outliers beyond those values are shown by solid dots.}}

[[Figures 5-9: Intact rock glacier density maps would paint a better picture than plotting centroids. Please reconsider the actual value of the current figures 5-9.]]

{{We are uncertain what you mean by this comment. Original Figures 5 through 8 (updated figures 6 through9) as originally presented absolutely do show both the locations of individual rock glaciers, as well as the class centroids.}}

[[Table 1: The inventories by Kellerer-Pirklbauer et al (2012) and Krainer and Ribis (2012) are in the meantime replaced by a consistent rock glacier inventory of Austria (Wagner et al., 2020); see notes of referee #1 about this inventory and the one available for the Balkan Peninsula by Magori et al (2020). Moreover, the table would be of greater value if regions would be mentioned; e.g. Seppi et al. (2012): Eastern Italian Alps (Trentino). ]]

{{Thank you for the suggestion, regions have been added to the table in the revised

manuscript.}}

Please also note the supplement to this comment:
https://essd.copernicus.org/preprints/essd-2020-158/essd-2020-158-AC1-supplement.pdf

---

## Referee Report (RR1)

740

[referee-annotated manuscript omitted]

---

## Referee Report (RR2)

Referee comments in [[DOUBLE BRACKETS]], author responses in {{DOUBLE BRACES}}.

**[[Anonymous Referee #1**

General Comments: This contribution presents a nation-wide inventory of "intact" rock glaciers (sensu Barsch, 1996) and fully mantled debris-covered glaciers for the contiguous USA. The topic is suitable to ESSD. The authors justify their work with the need for continental-scale inventories, which are currently not available. This is clearly an impressive mapping effort. On the down side, I find the mapping rules adopted, the inherent mapping uncertainty and the metadata specifics to be insufficiently illustrated (i.e., with figures and photos) and documented. I also note a number of drawbacks in the inventorying approach that need to be considered carefully, before this database maybe considered for further analysis. In particular, the typology of landforms (i.e., intact rock glaciers and fully mantled debris-covered glaciers) blended in and the dynamic classification scheme adopted, make the present inventory not comparable with other existing inventories around the world. For these reasons, making statistical inference from this database in its present form may lead to misleading conclusions.

Major points to be addressed are summarized under the following headings:
1. Rock glaciers and debris-covered glaciers: Although I agree that making a clearcut distinction between rock glaciers and debris-covered glaciers in some cases is subject to large uncertainties, which can only be resolved with direct geophysical investigation, a number of morphological features are known to be distinctive of debris-covered glaciers. These include, but are not limited to, the presence of crevasses with exposed ice, ice cliffs, abundant thermokarst and supraglacial lakes, supraglacial streams, outflow breaches. In this regard, adding some sample images showing which kind of debris-covered glaciers were excluded from the inventory would help the reader a lot. I suggest that the authors add a field in the PSURGI attribute table indicating whether a given polygon is a rock glacier, a debris-covered glacier, or uncertain i.e., when they are unable to distinguish between the two.]]

{{Thank you for highlighting an ambiguity be believed was clear; extensive text has been added to the revised manuscript to reassure readers that very few, if any, debriscovered glaciers are likely to have been inadvertently included. All features were visually reviewed once more and 11 were deleted from the inventory as being likely debris-covered glaciers. Additionally, we have adopted a clear distinction throughout that the focus of this inventory is active rock glaciers. Of the features distinctive of debris-covered glaciers you list, none were observed on features that had not already been excluded from the inventory for having expansive bare glacial ice in their accumulation zones. Original manuscript text clearly identifies "fully mantled debriscovered glaciers" as those which may inadvertently be included in the inventory, and clarifies that we followed previous studies and "omit features with expansive bare glacial ice in their accumulation zones as those are clearly debris-covered glaciers" (section 2.2). Additionally, and by definition, any debris-covered glacier that is "fully mantled" with regolith would not appear to have any crevasses with exposed ice or ice cliffs visible in all but the very highest resolution satellite imagery (i.e., sub-meter resolution) which is not widely and freely available, and was not used to create this inventory. As no discrimination between "fully mantled debris-covered glaciers" inadvertently included (i.e., fully mantled debris-covered glaciers that lack expansive surfaces of exposed ice in their accumulation zones or obvious supraglacial}}

lakes/streams) can be confidently made though surface analysis alone given data limitations, no additional attribute data will be added. The operative term in describing how we addressed "ambiguous" debris covered glaciers is "fully mantled", and hopefully that is made clear in the revised text.}}

*I am happy with such changes, thank you for addressing this point in the revised manuscript.*

[[2. Degree of activity classification scheme: The dynamic classification scheme adopted in PSURGI subdivides intact rock glaciers into three classes: highly, intermediately, and minimally active. This approach makes PSURGI not immediately comparable with most of existing inventories around the world, which discriminate intact rock glaciers into inactive (i.e., no front movement) and active landforms (Barsch, 1996).
Recent mapping tests in Northern Tyrol have shown that distinction between active and inactive rock glaciers is subject to high uncertainty, and that inactive rock glaciers (those that supposedly should move more slowly) displayed large disagreement among a pool of international, experienced mappers (Brardinoni et al., 2019). In this context, subdividing intact rock glaciers into three categories (as opposed to the classical two) appears unreliable. Along these lines, PSURGI approach to dynamic classification seems contradictory: on one hand it is stated that visual interpretation of imagery does not afford distinction between rock glaciers and debris-covered glaciers, on the other hand, this same procedure would allow to discriminate three subtypes of intact rock glaciers. I believe that this type of fine distinction could be achieved reliably only with the aid of InSAR technology. For the reasons outlined above, I suggest that the authors revert their dynamic classification scheme for intact rock glaciers to the classical one (i.e., active and inactive).]]

{{We believe this inventory can indeed be readily and directly compared with any other inventories that identify "active" rock glaciers, provided those inventories also took the care to either provide an "active/inactive" attribute in any spatial data made available, or discriminate "active/inactive" features in their statistical analysis. We believe all features included in the inventory to fit accurately be described as "active" (i.e., exhibiting at least some flow annually), and are unaware of a universally accepted non-zero movement threshold that discriminates "active" form "inactive" rock glaciers.

*It is important that the authors define what they mean by active, inactive and relict rock glaciers, providing reference to papers where such definitions are formulated. By far, the international reference for the qualitative visual classification of rock glacier degree of activity in inventories remains that consolidated in Barsch (1996). In there, authors will find what is meant by active (e.g., front downslope movement), inactive (no detectable front movement but vertical and/or downslope deformation possible on other parts of the rock glacier) and relict (no detectable deformation) rock glaciers, including thresholds of detectable motion, as well as vertical (e.g., subsidence) or downslope deformation (e.g., front motion). If the authors are not happy with Barsch (1996), they should provide alternative references applicable to regional, remotely-derived rock glacier inventories.*

*Please consider that the foremost application of rock glacier inventories lies in their ability to provide independent information on the spatial distribution of mountain permafrost, to test and refine existing permafrost maps or build new ones (e.g., Boeckli et al., 2012; Schmid et al., 2015). These studies, which rely on Barsch's classification scheme, subdivide rock glaciers into intact (active and inactive) and relict. The former landforms*

*suggest local permafrost occurrence and the latter exclude it. Following this logic, an inventory including active rock glaciers only, would not be able to serve this permafrost-oriented goal. This is fine, as long as the authors are willing to mention this limitation and consider adding reference to:*
*Boeckli L, Brenning A, Gruber S, Noetzli J. 2012. Permafrost distribution in the European Alps: calculation and evaluation of an index map and summary statistics. The Cryosphere 6(4): 807–820.*

*Schmid MO, Baral P, Gruber S, Shahi S, Shrestha T, Stumm D, Wester P. 2015. Assessment of permafrost distribution maps in the Hindu Kush Himalayan region using rock glaciers mapped in Google Earth. The Cryosphere 9(6): 2089–2099.*

Numerous widely-cited rock glacier inventories use a variety of classification systems containing anywhere from six classes to a single class; two classes is by no means a global standard, and among those that do use two classes the specific class definitions and classification rules occasionally vary to a degree precluding direct comparison from one inventory to another.
We agree InSAR or equivalent technology would be necessary if our classification system was quantitative and attempted to estimate actual flow rates, but the three-class system we employ is purely qualitative, based primarily on prevalence of ridge and swale surface banding and oversteepened terminal and lateral slopes. Ridge and swale surface banding is widely accepted to be the result of differential flow rates (i.e., absent from rock glaciers that are not presently or recently flowing), while oversteepened terminal and lateral slopes are widely accepted to be the result of cementing by flow motivated by the deformation of interstitial ice. We agree that there is much uncertainty and disagreement with respect to classifying rock glaciers, and state as much numerous times throughout the paper, but believe that our three-class system offers valuable flexibility to rock glacier researchers. Those more interested in active rock glaciers exhibiting the highest degree of classification certainty can focus on Class 1 features, while those more interested in probably currently active, though possibly recently inactive, rock glaciers and willing to accept a slightly lower degree of classification certainty can focus on Class 3 features.}}

[[3. Completeness: The question of inventory completeness is only brushed upon. A similar large-scale inventory should be coupled by a systematic testing on the variability and uncertainty among mappers involved in the inventory. For example, Google Earth Imagery, when not complemented by LiDAR-derived hillshades and high-resolution orthophoto mosaics has been shown to yield incomplete rock glacier detection, especially due to poor distinction between adjacent coalescing lobes (Brardinoni et al., 2019). In this context, the question of complex multi-lobe (or polymorphic) rock glaciers and the way in which these morphologies were mapped is not addressed. No example was provided. Any geomorphologist familiar with rock glacier mapping is aware of the inherent uncertainties associated with an inventory, yet the authors depict PSURGI as greatly accurate. Please consider tuning down some sentences in that section.]]

{{As in the paper you reference, the systemic testing of classification variability we performed (Section 4.2) showed slightly less agreement among analysts from most active to least active features, but just as in the paper you reference, the systemic testing of classification variability we performed showed broad agreement for all three feature classes. Additionally, our inventory was compared to three other smaller regional inventories and showed high levels of agreement with each of them. We agree LiDAR-derived DEMs would likely increase the accuracy of this inventory, but LiDAR is not available for the vast majority of the study area. Text has been added to the manuscript to describe how we addressed multi-lobate rock glaciers (i.e., we link distinct accumulation zones to distinct rock glaciers), though it is worth noting that there is little agreement among rock glacier researchers on which features to focus on when classifying multi-lobate rock glaciers, or exactly when two adjacent lobes with a common accumulation zone should be considered two distinct rock glaciers. We are uncertain exactly which section you are referring to in your request that we "tune down some sentences in that section", but both the comparisons to other regional inventories and the comparisons to classifications between analysts support the conclusion that our active rock glacier inventory is indeed quite accurate. Regardless, additional text has been added to revised manuscript to further clarify uncertainties and limitations.}}

*Thank you for adding more text on this point. As per previous comment, clarifying what is intended by active, inactive and relict will help comparison with prior studies conducted in section 4.2 (i.e., Janke, 2007; Millar and Westfall, 2008; Liu et al., 2013).*

*In lines 348-350, the authors state:*
*"The 2013 California study (Liu et al., 2013) reported 67 "active" rock glaciers, a subset of features identified in the 2008 study and the category in that study most similar to our Class 1 classification criteria, while we identified 88 active rock glaciers in largely the same study region."*

*I see a count difference of 21 units between PSUARGI and those mapped by Liu et al., 2013. Please clarify.*

*As per prior comment, making direct reference to existing studies that have documented operator-based uncertainty will be useful for providing broader international context to section 4.2.*

[[4. Rock glacier delineation (mapping rules): No specific description of the mapping rules applied in PSURGI is provided, and only vague wordy descriptions are given. For example, one of the most problematic issues when delineating a rock glacier polygon is

typically represented by the extent of the rooting zone, which borders the upper end of a rock glacier. In the manuscript, I could not find which mapping rule has been applied to delineate the upper end of rock glaciers and exclude the rooting zone (assuming this was excluded from the mapped polygons). In Figure 1, class 2 example, the upper end of the polygon cuts across flow lines, following no apparent discontinuity in curvature or roughness. Was the mapping confidence consistent across the entire perimeter of this polygon? Overall, the three examples provided in Figure 1 do not struck me for being indicative of accurate mapping. Please add more examples and/or refine the outlines of the current ones.]]

{{Since the three-class system we employ is purely qualitative, the mapping rules are also qualitatively described (Section 2.2). Despite your characterization of these mapping rules as "vague", these were the same rules described verbatim to the five independent analysts who systemically, and in isolation, tested our classification rules, which resulted in broad agreement between analysts for all three classes. With respect to rooting zone extents, we focused on sharp changes in slope, "from the steep slopes of exposed bedrock and unconsolidated talus in the rock glacier accumulation zone to the more gentle slope of the main body of the ice-thickened rock glacier". Text has been added to the manuscript to describe identification of upper rock glacier boundaries where exposed bedrock was not present as less confidently delineated than sharp lateral and terminal boundaries. The Class 2 example boundary in original Figure 1 (updated to Figure 2) does not cross any flow lines. Copious flow lines, which generally trend perpendicular to the fall line, are visible at the lower end of the rock glacier, but none are visible at the upper end. These three examples are broadly representative of the three inventory class delineations overall and will not be edited or replaced.}}

*Up to the authors. Any reader will be able to make her/his own personal judgement on outlining accuracy based on the examples provided in Figure 2. In both class-2 examples provided, the polygon outline cuts across flow lines.*
*Flow lines in a rock glacier are those indicating the (current or former) downward direction of debris transfer operated by creep, from the rooting zone towards the front. Lines transversal to the flow form the so-called ridge and furrow topography.*

[[5. Metedata: A database submitted for publication should come with well-documented metadata, including; i) A list of attributes in table format (i.e., the attribute table in the shapefile includes the dynamic classification only). ii) a list of complementary imagery used other than Google Earth Pro (i.e., currently the authors state the following in lines 83-85: "... supplementing with other plan-view imagery imported into ArcMap 10.4 when Google Earth Pro imagery was unsuitable due to cloud cover or other issues."]]

{{Shapefile attributes have been added, and a descriptive attribute table has been added to the revised manuscript. Citation of additional plan-view imagery used has been added to the revised manuscript.}}

*I think this was a key limitation of the original submission. Thank you for making available a complete list of rock glacier attribute.*

[[Specific Comments:
Title: considering the nature of the inventory, the title of the paper should acknowledge the inclusion of (intact) rock glaciers and fully mantled debris-covered glaciers.]]

{{Manuscript title, and text throughout revised manuscript, has been updated to reflect our focus on active rock glaciers. Considering the original text explained omission of "features with expansive bare glacial ice in their accumulation zones as those are clearly debris-covered glaciers", and the lack of supraglacial lakes or streams observed on remaining features retained, we are confident that very few "fully mantled debriscovered glaciers" were included in the inventory. Regardless, extensive additional text added throughout the revised manuscript to make our attempts to exclude "fully mantled debris-covered glaciers" even more explicit. Referencing "debris-covered" glaciers in the title when the relevant caveats are discussed numerous times in the manuscript does little to assist the intended audience, and we are confident that very few debris-covered glaciers are likely to have been inadvertently included. Given no further discrimination between the "fully mantled debris-covered glaciers" that may have been inadvertently included (i.e., those with no visible crevasses, ice walls, exposed glacial ice, or supraglacial lakes/streams) and active rock glaciers can be confidently made through surface analysis alone given data limitations, and no attribute data will be added.}}

*Thank you for addressing this question.*

[[Lines 23-25: "Two lesser known components of the montane cryosphere are rock glaciers and debris-covered glaciers, though presently there are no widely accepted formal definitions of either feature type that can be used to universally and unambiguously discriminate the two for all purposes"
In my opinion, this approach leads to confusion. The statement is not supported by any reference and discounts decades of research focused respectively on debris-covered glaciers and rock glaciers. It is more fair to say that there are widely accepted definitions of rock glaciers and debris-covered glaciers, and that a minority disagrees.]]

{{Extensive text has been added to revised manuscript to describe the subset of debris covered glaciers (i.e., those with no visible crevasses, ice walls, exposed glacial ice, or supraglacial lakes/streams) that may have inadvertently been included due to our inability to discriminate them from active rock glaciers based on aerial/satellite imagery alone. Text and citations have been added to the manuscript to briefly describe the widely accepted "continuum concept" of glaciers, rock glaciers and debris-covered glaciers, as well as the common transition of debris-covered glaciers to rock glaciers.
We will have to agree to disagree about how widely accepted your preferred definitions of debris-covered glaciers and rock glaciers are since you do not provide them in your comments here, but we are unaware of any rock glacier researchers who reject the "continuum concept" as wholly without merit. We agree that distinctions occasionally can, and when possible should, be made between rock glaciers and "fully mantled debris-covered glaciers" (i.e., those with no visible crevasses, ice walls, exposed glacial ice, or supraglacial lakes/streams), but such distinctions can only be confidently made with detailed field surveys (e.g., coring, GPR, etc.), cannot be made using manual aerial/satellite image classification, will ultimately come down to semantics based on internal ice fraction and arrangement, and should not be a barrier to disseminating this active rock glacier inventory to the rock glacier research community.}}

*I did provide reference to definitions on rock glaciers by referencing to Barsch (1996), and follow up papers (i.e., Haeberli et al., 2006). I also did provide features that are known for differentiating debris-covered glaciers from rock glaciers in one of my other comments. Please consider having a look at the website of the IPA (International Permafrost Association) action group on rock glacier inventories, hosted by the University of Fribourg.*

*In there you will find Baseline concepts and guidelines, including updated definitions of rock glaciers and morphological differences with debris-covered glaciers.*

*https://www.unifr.ch/geo/geomorphology/en/research/ipa-action-group-rock-glacier/*

[[Lines 37-38: "Fully debris-covered glaciers are indistinguishable from the more traditionally defined rock glaciers through surface analysis alone".
Please try to support similar clearcut statements with references and illustrative examples (i.e., figures).]]

{{We have added considerable clarifying/qualifying text to the passage you referenced, but the original point remains relevant and unchanged: fully mantled debris-covered glaciers (i.e., those with no visible crevasses, ice walls, exposed glacial ice, or supraglacial lakes/streams) are, in most cases, very difficult, if not impossible, to confidently discriminate from active rock glaciers through surface analysis alone.}}

*Thank you for addressing this point.*

[[Lines 41-42: "The semantics of classifying these two cryospheric feature types is occasionally debated, but is not something we seek to resolve with this inventory (Clark et al. 1998, Potter 1972, Haeberli et al. 2006, Berthling 2011)."
I think the question is way beyond semantics. I understand that you do not want to enter into this dispute, but there are a number of morphological attributes that in many instances should aid guiding distinction between rock glaciers and debris-covered glaciers. Merging rock glaciers and debris-covered glaciers in one database without any sort of morphological distinction represents a major limitation of this inventory.]]

{{We appreciate you comment, but fear you have drastically overestimated the number of debris covered glaciers that have been inadvertently included in our active rock glacier inventory. See our numerous responses above referencing imagery limitations with respect to confidently discriminating all "fully mantled debris-covered glaciers" (i.e., those with no visible crevasses, ice walls, exposed glacial ice, or supraglacial lakes/streams) from active rock glaciers, as well as extensive text expansion on this topic in the introduction of the revised manuscript. Again, we are confident that very few debris covered glaciers were included in the active rock glacier inventory.}}

*Thank you for clarifying this point. In the manuscript, there was no quantitative information through which I could guess the proportion of inadvertently misclassified rock glaciers.*

[[Lines 131-134: "To partially address this ambiguity all features identified as rock glaciers were subsequently assigned to a three-tier classification system based on surface characteristics known to correlate with downslope movement motivated by deformation of the internal ice-rock matrix (Figure 1)."
Which would be the surface characteristics known to correlate with downslope movement? Please provide empirical data or reference to empirical publications showing such correlations. My impression is that by increasing the number of activity classes (three in this case), one is going to increase the degree of uncertainty. Please see my general comment #2.]]

{{Ridge and swale surface banding, commonly referred to as "flow banding", is widely accepted to be the result of differential flow rates and is highlighted in virtually every peer

reviewed publication that focuses on rock glacier flow rates. We believe this correlation is patently obvious to any rock glacier researchers, our intended audience, but have added several citations to Section 2.2 to address the reviewer's concern. More flow banding indicates more flow, which in turn indicates higher levels of rock glacier activity. We are confident that all features included in the inventory can accurately be described as "active", and by providing additional qualitative information in the form of our our three-tired classification scheme we have added flexibility to future applications of the inventory. As demonstrated in our blind tests of classifications completed by numerous analysts, classification homogeneity is high across all three classes, but highest for Class 1 features. If readers wish to further analyze the inventory and only include those features with the very highest classification confidence, they can focus on Class 1 features. If readers wish to further analyze the inventory and are satisfied with slightly lower classification confidence, they can include all three classes of features.}}

*I thank the authors for their reply but I fear this is not going to solve the question I raised. This three-class classification scheme for active rock glaciers tends to read too much solely from visual inspection of optical imagery. Apparently, it makes difficult direct comparison with existing studies i.e., see last part of section 4.2, where the authors have to guess which PSUARGI class (1, 2 or 3) to consider against.*
*For example, why do they compare PSUARGI Class 1 with "Active" rock glaciers in Janke (2008)? If PSUARGI included active rock glaciers only, then Janke's mapping should be compared with the combination of classes 1, 2 and 3.*

*As mentioned earlier in this review, without a better definition of how PSUARGI active rock glaciers compare to the three-part classical classification (active, inactive and relict), this inventory remains difficult to be considered in the broader international context.*

[[Line 155: "after removing 146 small (< 0.01 km2) Class 3 rock glaciers following glaciological convention of area thresholds".
Why using this threshold size? Rock glaciers are not glaciers, neither are included in the World Glacier Inventory.]]

{{Rock glacier research is very much informed by glacier research, a discipline with much more robust inventories available due to much more robust remote sensing analysis techniques available, mostly stemming from the spectral reluctance of exposed ice. Glacier inventories have identified a lower area threshold beyond which glaciers cannot be confidently identified and delineated, and given rock glaciers are much more difficult to identify than glaciers it seems only prudent that we apply the same area threshold. In any event, the omitted features would have contributed only a minuscule fraction of the features included by count, and a virtually infinitesimal fraction of the features included by area.}}

*The question is not as straightforward as pictured by the authors. The WGI is largely based on satellite imagery, and glaciers are much more dynamic in nature than rock glaciers. As such, their mapping include uncertainty related to distinguishing between snow, ice and firn, which does not really apply to rock glaciers. Anyway, my question was simple minded and I was just asking to write a sentence in which you justify your choice for such a threshold size.*

[[Conclusions (lines 252-260): most of the conclusions paragraph reads more like introduction. Please consider rewriting and connecting the conclusions to the main results outlined in the manuscript.]]
{{Conclusions section in revised manuscript has been expanded and refocused as you suggest.}}

*Thank you for rewriting the conclusions.*

[[List of existing inventories:
I suggest adding the following references to the list of existing inventories, the first includes 5769 rock glaciers across Austria:
Wagner et al 2020. The first consistent inventory of rock glaciers and their hydrological catchments of the austrian alps. Austrian Journal of Earth Sciences, 113: 1-23.
Brigitte Magori, Petru Urdea, Alexandru Onaca & Florina Ardelean (2020) Distribution and characteristics of rock glaciers in the Balkan Peninsula, Geografiska Annaler:
Series A, Physical Geography, DOI: 10.1080/04353676.2020.1809905]]

{{Thank you for the suggestion, both references have been added to the revised manuscript.}}

---

## Author Response (AR3)

*Final Comments to the Author, with responses in red:*

Dear Gunnar et al, many thanks for the thorough revision of two review rounds, I think that the paper is now almost ready to be published. There are only some minor points to be addressed:

(1) please include a copy of your table 5 "Portland State University Active Rock Glacier Inventory shapefile attribute data dictionary" in the data publication at PANGAEA. These kind of definitions must be also available with the data

Added data dictionary to Pangaea data repository.

(2) please contact PANGAEA and ask them to register the DOI after they have added the table with the variable definitions to the repository. At the moment, the data are still "in review". For a final acceptance we need a registered DOI. Please also change all "https://doi.pangaea.de" links to "https://doi.org" links (in the text and references section)

DOI is now fully registered.

(3) (line 265 in the track-change mode version) I would be further pleased if you followed the wish of the reviewer to add some words explaining why you chose the same threshold like for glaciers (i.e. you could simply use your argumentation on page 10 of the authors answers in orange, or add "as a minimum approach"). It would be great if the reader doesn't need to ask the same question like the reviewer, but intuitively understand your rationale.

Added sentence clarifying all rationale for excluding very small (<0.01 km$^2$) rock glaciers.

many thanks and best regards,

Kirsten Elger

*Second round of Referee's comments are written in green Italics, second round of author responses in orange italics.*

First round of Referee's comments in [[DOUBLE BRACKETS]], first round of author responses in {{DOUBLE BRACES}}.

**[[Anonymous Referee #1**

General Comments: This contribution presents a nation-wide inventory of "intact" rock glaciers (sensu Barsch, 1996) and fully mantled debris-covered glaciers for the contiguous USA. The topic is suitable to ESSD. The authors justify their work with the need for continental-scale inventories, which are currently not available. This is clearly an impressive mapping effort. On the down side, I find the mapping rules adopted, the inherent mapping uncertainty and the metadata specifics to be insufficiently illustrated (i.e., with figures and photos) and documented. I also note a number of drawbacks in the inventorying approach that need to be considered carefully, before this database maybe considered for further analysis. In particular, the typology of landforms (i.e., intact rock glaciers and fully mantled debris-covered glaciers) blended in and the dynamic classification scheme adopted, make the present inventory not comparable with other existing inventories around the world. For these reasons, making statistical inference from this database in its present form may lead to misleading conclusions.

Major points to be addressed are summarized under the following headings:

1. Rock glaciers and debris-covered glaciers: Although I agree that making a clearcut distinction between rock glaciers and debris-covered glaciers in some cases is subject to large uncertainties, which can only be resolved with direct geophysical investigation, a number of morphological features are known to be distinctive of debris-covered glaciers. These include, but are not limited to, the presence of crevasses with exposed ice, ice cliffs, abundant thermokarst and supraglacial lakes, supraglacial streams, outflow breaches. In this regard, adding some sample images showing which kind of debris-covered glaciers were excluded from the inventory would help the reader a lot. I suggest that the authors add a field in the PSURGI attribute table indicating whether a given polygon is a rock glacier, a debris-covered glacier, or uncertain i.e., when they are unable to distinguish between the two.]]

{{Thank you for highlighting an ambiguity be believed was clear; extensive text has been added to the revised manuscript to reassure readers that very few, if any, debriscovered glaciers are likely to have been inadvertently included. All features were visually reviewed once more and 11 were deleted from the inventory as being likely debris-covered glaciers. Additionally, we have adopted a clear distinction throughout that the focus of this inventory is active rock glaciers. Of the features distinctive of debris-covered glaciers you list, none were observed on features that had not already been excluded from the inventory for having expansive bare glacial ice in their accumulation zones. Original manuscript text clearly identifies "fully mantled debriscovered glaciers" as those which may inadvertently be included in the inventory, and clarifies that we followed previous studies and "omit features with expansive bare glacial ice in their accumulation zones as those are clearly debris-covered glaciers" (section 2.2). Additionally, and by definition, any debris-covered glacier that is "fully mantled" with regolith would not appear to have any crevasses with exposed ice or ice cliffs visible in all but the very highest resolution satellite imagery (i.e., sub-meter resolution) which is not widely and freely available, and was not used to create this inventory. As no discrimination between "fully mantled debris-covered glaciers" inadvertently included (i.e., fully mantled debris-covered glaciers that lack expansive surfaces of exposed ice in their accumulation zones or obvious supraglacial lakes/streams) can be confidently made though surface analysis alone given data limitations, no additional attribute data will be added. The operative term in describing how we addressed "ambiguous" debris covered glaciers is "fully mantled", and hopefully that is made clear in the revised text.}}

*I am happy with such changes, thank you for addressing this point in the revised manuscript.*

[[2. Degree of activity classification scheme: The dynamic classification scheme adopted in PSURGI subdivides intact rock glaciers into three classes: highly, intermediately, and minimally active. This approach makes PSURGI not immediately comparable with most of existing inventories around the world, which discriminate intact rock glaciers into inactive (i.e., no front movement) and active landforms (Barsch, 1996).
Recent mapping tests in Northern Tyrol have shown that distinction between active and inactive rock glaciers is subject to high uncertainty, and that inactive rock glaciers (those that supposedly should move more slowly) displayed large disagreement among a pool of international, experienced mappers (Brardinoni et al., 2019). In this context, subdividing intact rock glaciers into three categories (as opposed to the classical two) appears unreliable. Along these lines, PSURGI approach to dynamic classification seems contradictory: on one hand it is stated that visual interpretation of imagery does not afford distinction between rock glaciers and debris-covered glaciers, on the other hand, this

same procedure would allow to discriminate three subtypes of intact rock glaciers. I believe that this type of fine distinction could be achieved reliably only with the aid of InSAR technology. For the reasons outlined above, I suggest that the authors revert their dynamic classification scheme for intact rock glaciers to the classical one (i.e., active and inactive).]]

{{We believe this inventory can indeed be readily and directly compared with any other inventories that identify "active" rock glaciers, provided those inventories also took the care to either provide an "active/inactive" attribute in any spatial data made available, or discriminate "active/inactive" features in their statistical analysis. We believe all features included in the inventory to accurately be described as "active" (i.e., exhibiting at least some flow annually), and are unaware of a universally accepted non-zero movement threshold that discriminates "active" from "inactive" rock glaciers.

*It is important that the authors define what they mean by active, inactive and relict rock glaciers, providing reference to papers where such definitions are formulated. By far, the international reference for the qualitative visual classification of rock glacier degree of activity in inventories remains that consolidated in Barsch (1996). In there, authors will find what is meant by active (e.g., front downslope movement), inactive (no detectable front movement but vertical and/or downslope deformation possible on other parts of the rock glacier) and relict (no detectable deformation) rock glaciers, including thresholds of detectable motion, as well as vertical (e.g., subsidence) or downslope deformation (e.g., front motion). If the authors are not happy with Barsch (1996), they should provide alternative references applicable to regional, remotely-derived rock glacier inventories.*

*Please consider that the foremost application of rock glacier inventories lies in their ability to provide independent information on the spatial distribution of mountain permafrost, to test and refine existing permafrost maps or build new ones (e.g., Boeckli et al., 2012; Schmid et al., 2015). These studies, which rely on Barsch's classification scheme, subdivide rock glaciers into intact (active and inactive) and relict. The former landforms suggest local permafrost occurrence and the latter exclude it. Following this logic, an inventory including active rock glaciers only, would not be able to serve this permafrost- oriented goal. This is fine, as long as the authors are willing to mention this limitation and consider adding reference to:*
*Boeckli L, Brenning A, Gruber S, Noetzli J. 2012. Permafrost distribution in the European Alps: calculation and evaluation of an index map and summary statistics. The Cryosphere 6(4): 807–820.*

*Schmid MO, Baral P, Gruber S, Shahi S, Shrestha T, Stumm D, Wester P. 2015. Assessment of permafrost distribution maps in the Hindu Kush Himalayan region using rock glaciers mapped in Google Earth. The Cryosphere 9(6): 2089–2099.*

*Thank you for the suggested references, all have been added. We have identified the definitions proposed by Barsch (1996) as those we employ and agree with, and have made specific reference to limitations of applying our active rock glacier inventory to validating permafrost area models as in Boeckli et al. 2012 and Schmid et al. 2015.*

Numerous widely-cited rock glacier inventories use a variety of classification systems containing anywhere from six classes to a single class; two classes is by no means a global standard, and among those that do use two classes the specific class definitions and classification rules occasionally vary to a degree precluding direct comparison from

one inventory to another. We agree InSAR or equivalent technology would be necessary if our classification system was quantitative and attempted to estimate actual flow rates, but the three-class system we employ is purely qualitative, based primarily on prevalence of ridge and swale surface banding and oversteepened terminal and lateral slopes. Ridge and swale surface banding is widely accepted to be the result of differential flow rates (i.e., absent from rock glaciers that are not presently or recently flowing), while oversteepened terminal and lateral slopes are widely accepted to be the result of cementing by flow motivated by the deformation of interstitial ice. We agree that there is much uncertainty and disagreement with respect to classifying rock glaciers, and state as much numerous times throughout the paper, but believe that our three-class system offers valuable flexibility to rock glacier researchers. Those more interested in active rock glaciers exhibiting the highest degree of classification certainty can focus on Class 1 features, while those more interested in probably currently active, though possibly recently inactive, rock glaciers and willing to accept a slightly lower degree of classification certainty can focus on Class 3 features.}}

*The new paragraph added in the introduction (lines 71-87) does not solve the classification question raised in the first round of review. In the new text, the term relict was replaced with inactive, as if these two were synonyms (i.e., line 72, but also in the last part of section 4.2: line 347). This is going to create confusion in the international audience.*
*Please state that PSUARGI does not include, for as much as possible, relict and inactive rock glaciers. In this context, it is important to corroborate inherent uncertainty in this kind of visual-based differentiation by referring to previous studies that have quantitatively evaluated and discussed this point, for example:*

*Schmid MO, Baral P, Gruber S, Shahi S, Shrestha T, Stumm D, Wester P. 2015. Assessment of permafrost distribution maps in the Hindu Kush Himalayan region using rock glaciers mapped in Google Earth. The Cryosphere 9(6): 2089–2099.*

*Brardinoni F, Scotti R, Sailer R, and Mair V. 2019. Sources of uncertainty and variability in rock glacier inventories. Earth Surface Processes and Landforms, 44, 2450-2466.*

*On the question of PSUARGI classification of active rock glaciers into three sub-classes, please see also reply to detailed comment further down in this review.*

*Thank you for the suggested references, all have been added. Throughout the manuscript we have repeatedly stressed that we only attempted to inventory active rock glaciers, and deliberately sought to exclude both inactive and relict rock glaciers. We have made very clear what definition of "active" we employ, and exactly how we identified them based on specific features not generally present on inactive or relict rock glaciers. As all rock glacier researchers well understand, no two image analysts will ever perfectly agree in their interpretation of aerial imagery and subsequent classification of rock glaciers identified. We have repeatedly stressed that certainty in rock glacier classification can only be attained through detailed field investigation, but have laid out a clear methodology that agrees with numerous other inventories for discriminating active rock glaciers from both inactive and relict rock glaciers. As you well know, and as in many of the references you have suggested and we have included also attest, discriminating between inactive and relict rock glaciers from aerial image classification alone is virtually impossible. Given the clarity and specificity of the characteristics of the active rock glaciers included in the inventory, and the numerous direct statements that we deliberately exclude inactive and relict rock glaciers, we do not see*

*the need to add copious additional text defining types of rock glaciers explicitly not included in the inventory. Readers of this paper, and those who use the geospatial inventory, will not be confused by any inadvertent conflation of inactive and relict rock glacier definitions because the inventory quite explicitly only includes well defined, and confidently identifed, active rock glaciers.*

[[3. Completeness: The question of inventory completeness is only brushed upon. A similar large-scale inventory should be coupled by a systematic testing on the variability and uncertainty among mappers involved in the inventory. For example, Google Earth Imagery, when not complemented by LiDAR-derived hillshades and high-resolution orthophoto mosaics has been shown to yield incomplete rock glacier detection, especially due to poor distinction between adjacent coalescing lobes (Brardinoni et al., 2019). In this context, the question of complex multi-lobe (or polymorphic) rock glaciers and the way in which these morphologies were mapped is not addressed. No example was provided. Any geomorphologist familiar with rock glacier mapping is aware of the inherent uncertainties associated with an inventory, yet the authors depict PSURGI as greatly accurate. Please consider tuning down some sentences in that section.]]

{{As in the paper you reference, the systemic testing of classification variability we performed (Section 4.2) showed slightly less agreement among analysts from most active to least active features, but just as in the paper you reference, the systemic testing of classification variability we performed showed broad agreement for all three feature classes. Additionally, our inventory was compared to three other smaller regional inventories and showed high levels of agreement with each of them. We agree LiDAR-derived DEMs would likely increase the accuracy of this inventory, but LiDAR is not available for the vast majority of the study area. Text has been added to the manuscript to describe how we addressed multi-lobate rock glaciers (i.e., we link distinct accumulation zones to distinct rock glaciers), though it is worth noting that there is little agreement among rock glacier researchers on which features to focus on when classifying multi-lobate rock glaciers, or exactly when two adjacent lobes with a common accumulation zone should be considered two distinct rock glaciers. We are uncertain exactly which section you are referring to in your request that we "tune down some sentences in that section", but both the comparisons to other regional inventories and the comparisons to classifications between analysts support the conclusion that our active rock glacier inventory is indeed quite accurate. Regardless, additional text has been added to revised manuscript to further clarify uncertainties and limitations.}}

*Thank you for adding more text on this point. As per previous comment, clarifying what is intended by active, inactive and relict will help comparison with prior studies conducted in section 4.2 (i.e., Janke, 2007; Millar and Westfall, 2008; Liu et al., 2013).*

*In lines 348-350, the authors state:*
*"The 2013 California study (Liu et al., 2013) reported 67 "active" rock glaciers, a subset of features identified in the 2008 study and the category in that study most similar to our Class 1 classification criteria, while we identified 88 active rock glaciers in largely the same study region."*

*I see a count difference of 21 units between PSUARGI and those mapped by Liu et al., 2013. Please clarify.*

*As per prior comment, making direct reference to existing studies that have documented operator-based uncertainty will be useful for providing broader international context to section 4.2.*

*Like many other widely cited and well received rock glaciers inventories that do not use the active, inactive and relict classes, neither Liu et al. 2013 nor Millar and Westfall 2008 employed that classification scheme. Hopefully the IPA rock glacier action group, or some other body, will be able to provide the rock glacier research community with widely accepted guidelines for rock glacier inventories, but in recent decades through today, the active, inactive and relict classes are far from universally employed. As previously noted, numerous widely-cited rock glacier inventories use a variety of classification systems containing anywhere from six classes to a single class. With regards to comparison to Liu et al., we believe fewer rock glaciers were identified in that study because the main identification tool (InSAR) has numerous limitations based on speed, size and orientation of the primary rock glacier flow vector relative to the satellite sensor orientation, as described in that paper. We have added numerous citations to Brardinoni et al. 2019 to highlight uncertainty, as well as previous citations that also make the same point.*

[[4. Rock glacier delineation (mapping rules): No specific description of the mapping rules applied in PSURGI is provided, and only vague wordy descriptions are given. For example, one of the most problematic issues when delineating a rock glacier polygon is typically represented by the extent of the rooting zone, which borders the upper end of a rock glacier. In the manuscript, I could not find which mapping rule has been applied to delineate the upper end of rock glaciers and exclude the rooting zone (assuming this was excluded from the mapped polygons). In Figure 1, class 2 example, the upper end of the polygon cuts across flow lines, following no apparent discontinuity in curvature or roughness. Was the mapping confidence consistent across the entire perimeter of this polygon? Overall, the three examples provided in Figure 1 do not struck me for being indicative of accurate mapping. Please add more examples and/or refine the outlines of the current ones.]]

{{Since the three-class system we employ is purely qualitative, the mapping rules are also qualitatively described (Section 2.2). Despite your characterization of these mapping rules as "vague", these were the same rules described verbatim to the five independent analysts who systemically, and in isolation, tested our classification rules, which resulted in broad agreement between analysts for all three classes. With respect to rooting zone extents, we focused on sharp changes in slope, "from the steep slopes of exposed bedrock and unconsolidated talus in the rock glacier accumulation zone to the more gentle slope of the main body of the ice-thickened rock glacier". Text has been added to the manuscript to describe identification of upper rock glacier boundaries where exposed bedrock was not present as less confidently delineated than sharp lateral and terminal boundaries. The Class 2 example boundary in original Figure 1 (updated to Figure 2) does not cross any flow lines. Copious flow lines, which generally trend perpendicular to the fall line, are visible at the lower end of the rock glacier, but none are visible at the upper end. These three examples are broadly representative of the three inventory class delineations overall and will not be edited or replaced.}}

*Up to the authors. Any reader will be able to make her/his own personal judgement on outlining accuracy based on the examples provided in Figure 2. In both class-2 examples provided, the polygon outline cuts across flow lines.*

*Flow lines in a rock glacier are those indicating the (current or former) downward direction of debris transfer operated by creep, from the rooting zone towards the front. Lines transversal to the flow form the so-called ridge and furrow topography.*

*Again, we disagree that any of the polygon outlines in Figure 2 cross any flow lines, and absent specific annotations are unable to identify your assertion that they do. In Figure 2 the same rock glaciers and polygon outlines are shown in plan view (top) and oblique upslope view (bottom); there is only one example for each class. Assuming you are referencing talus to the north of the class 2 example (above the rock glacier in plan view, to the right of the rock glacier in oblique upslope view), it was deposited by rock slides, not by downslope creep.*

[[5. Metedata: A database submitted for publication should come with well-documented metadata, including; i) A list of attributes in table format (i.e., the attribute table in the shapefile includes the dynamic classification only). ii) a list of complementary imagery used other than Google Earth Pro (i.e., currently the authors state the following in lines 83-85: "... supplementing with other plan-view imagery imported into ArcMap 10.4 when Google Earth Pro imagery was unsuitable due to cloud cover or other issues."]]

{{Shapefile attributes have been added, and a descriptive attribute table has been added to the revised manuscript. Citation of additional plan-view imagery used has been added to the revised manuscript.}}

*I think this was a key limitation of the original submission. Thank you for making available a complete list of rock glacier attribute.*

[[Specific Comments:
Title: considering the nature of the inventory, the title of the paper should acknowledge the inclusion of (intact) rock glaciers and fully mantled debris-covered glaciers.]]

{{Manuscript title, and text throughout revised manuscript, has been updated to reflect our focus on active rock glaciers. Considering the original text explained omission of "features with expansive bare glacial ice in their accumulation zones as those are clearly debris-covered glaciers", and the lack of supraglacial lakes or streams observed on remaining features retained, we are confident that very few "fully mantled debriscovered glaciers" were included in the inventory. Regardless, extensive additional text added throughout the revised manuscript to make our attempts to exclude "fully mantled debris-covered glaciers" even more explicit. Referencing "debris-covered" glaciers in the title when the relevant caveats are discussed numerous times in the manuscript does little to assist the intended audience, and we are confident that very few debris-covered glaciers are likely to have been inadvertently included. Given no further discrimination between the "fully mantled debris-covered glaciers" that may have been inadvertently included (i.e., those with no visible crevasses, ice walls, exposed glacial ice, or supraglacial lakes/streams) and active rock glaciers can be confidently made through surface analysis alone given data limitations, and no attribute data will be added.}}

*Thank you for addressing this question.*

[[Lines 23-25: "Two lesser known components of the montane cryosphere are rock glaciers and debris-covered glaciers, though presently there are no widely accepted formal

definitions of either feature type that can be used to universally and unambiguously discriminate the two for all purposes"

In my opinion, this approach leads to confusion. The statement is not supported by any reference and discounts decades of research focused respectively on debris-covered glaciers and rock glaciers. It is more fair to say that there are widely accepted definitions of rock glaciers and debris-covered glaciers, and that a minority disagrees.]]

{{Extensive text has been added to revised manuscript to describe the subset of debris covered glaciers (i.e., those with no visible crevasses, ice walls, exposed glacial ice, or supraglacial lakes/streams) that may have inadvertently been included due to our inability to discriminate them from active rock glaciers based on aerial/satellite imagery alone. Text and citations have been added to the manuscript to briefly describe the widely accepted "continuum concept" of glaciers, rock glaciers and debris-covered glaciers, as well as the common transition of debris-covered glaciers to rock glaciers. We will have to agree to disagree about how widely accepted your preferred definitions of debris-covered glaciers and rock glaciers are since you do not provide them in your comments here, but we are unaware of any rock glacier researchers who reject the "continuum concept" as wholly without merit. We agree that distinctions occasionally can, and when possible should, be made between rock glaciers and "fully mantled debris-covered glaciers" (i.e., those with no visible crevasses, ice walls, exposed glacial ice, or supraglacial lakes/streams), but such distinctions can only be confidently made with detailed field surveys (e.g., coring, GPR, etc.), cannot be made using manual aerial/satellite image classification, will ultimately come down to semantics based on internal ice fraction and arrangement, and should not be a barrier to disseminating this active rock glacier inventory to the rock glacier research community.}}

*I did provide reference to definitions on rock glaciers by referencing to Barsch (1996), and follow up papers (i.e., Haeberli et al., 2006). I also did provide features that are known for differentiating debris-covered glaciers from rock glaciers in one of my other comments.*

*Please consider having a look at the website of the IPA (International Permafrost Association) action group on rock glacier inventories, hosted by the University of Fribourg. In there you will find Baseline concepts and guidelines, including updated definitions of rock glaciers and morphological differences with debris-covered glaciers.*

*https://www.unifr.ch/geo/geomorphology/en/research/ipa-action-group-rock-glacier/*

*We have included the Barsch reference at your suggestion (Haeberli 2006 was already cited several times), but cannot identify any meaningful or significant differences from definitions put forth in references you suggest and the definitions used in our last submission. We believe the detailed descriptions of what we consider to be an "active" rock glacier, and the steps employed to identify them and discriminate them from inactive and relict rock glaciers, agree well with the definitions used by the wider rock glacier research community. We appreciate the work of the IPA action group on rock glaciers and wish them success in their efforts to standardize rock glacier inventory methods, but hope you can see the value of our inventory which was completed well before the action group was ever convened.*

[[Lines 37-38: "Fully debris-covered glaciers are indistinguishable from the more traditionally defined rock glaciers through surface analysis alone".

Please try to support similar clearcut statements with references and illustrative examples (i.e., figures).]]

{{We have added considerable clarifying/qualifying text to the passage you referenced, but the original point remains relevant and unchanged: fully mantled debris-covered glaciers (i.e., those with no visible crevasses, ice walls, exposed glacial ice, or supraglacial lakes/streams) are, in most cases, very difficult, if not impossible, to confidently discriminate from active rock glaciers through surface analysis alone.}}

*Thank you for addressing this point.*

[[Lines 41-42: "The semantics of classifying these two cryospheric feature types is occasionally debated, but is not something we seek to resolve with this inventory (Clark et al. 1998, Potter 1972, Haeberli et al. 2006, Berthling 2011)."
I think the question is way beyond semantics. I understand that you do not want to enter into this dispute, but there are a number of morphological attributes that in many instances should aid guiding distinction between rock glaciers and debris-covered glaciers. Merging rock glaciers and debris-covered glaciers in one database without any sort of morphological distinction represents a major limitation of this inventory.]]

{{We appreciate you comment, but fear you have drastically overestimated the number of debris covered glaciers that have been inadvertently included in our active rock glacier inventory. See our numerous responses above referencing imagery limitations with respect to confidently discriminating all "fully mantled debris-covered glaciers" (i.e., those with no visible crevasses, ice walls, exposed glacial ice, or supraglacial lakes/streams) from active rock glaciers, as well as extensive text expansion on this topic in the introduction of the revised manuscript. Again, we are confident that very few debris covered glaciers were included in the active rock glacier inventory.}}

*Thank you for clarifying this point. In the manuscript, there was no quantitative information through which I could guess the proportion of inadvertently misclassified rock glaciers.*

[[Lines 131-134: "To partially address this ambiguity all features identified as rock glaciers were subsequently assigned to a three-tier classification system based on surface characteristics known to correlate with downslope movement motivated by deformation of the internal ice-rock matrix (Figure 1)."
Which would be the surface characteristics known to correlate with downslope movement? Please provide empirical data or reference to empirical publications showing such correlations. My impression is that by increasing the number of activity classes (three in this case), one is going to increase the degree of uncertainty. Please see my general comment #2.]]

{{Ridge and swale surface banding, commonly referred to as "flow banding", is widely accepted to be the result of differential flow rates and is highlighted in virtually every peer reviewed publication that focuses on rock glacier flow rates. We believe this correlation is patently obvious to any rock glacier researchers, our intended audience, but have added several citations to Section 2.2 to address the reviewer's concern. More flow banding indicates more flow, which in turn indicates higher levels of rock glacier activity. We are confident that all features included in the inventory can accurately be described as "active", and by providing additional qualitative information in the form of our our three-tired classification scheme we have added flexibility to future applications of the inventory. As demonstrated in our blind tests of classifications completed by numerous analysts,

classification homogeneity is high across all three classes, but highest for Class 1 features. If readers wish to further analyze the inventory and only include those features with the very highest classification confidence, they can focus on Class 1 features. If readers wish to further analyze the inventory and are satisfied with slightly lower classification confidence, they can include all three classes of features.}}

*I thank the authors for their reply but I fear this is not going to solve the question I raised. This three-class classification scheme for active rock glaciers tends to read too much solely from visual inspection of optical imagery. Apparently, it makes difficult direct comparison with existing studies i.e., see last part of section 4.2, where the authors have to guess which PSUARGI class (1, 2 or 3) to consider against.*
*For example, why do they compare PSUARGI Class 1 with "Active" rock glaciers in Janke (2008)? If PSUARGI included active rock glaciers only, then Janke's mapping should be compared with the combination of classes 1, 2 and 3.*

*As mentioned earlier in this review, without a better definition of how PSUARGI active rock glaciers compare to the three-part classical classification (active, inactive and relict), this inventory remains difficult to be considered in the broader international context.*

*Again, we assert that our three class system adds flexibility to future uses of the inventory as described in our initial response. We used our class 1 features when comparing to Janke 2008 because they are the class that was identified with methods that most directly accord with methods used by Janke to identify the "active" class. As described in our classification scheme, and a notion widely held in the rock glacier research community and described in numerous references cited, rock glacier activity is a continuum, not a simple binary. There will undoubtedly be some disagreement in the rock glacier research community on our classifications, as there are with classifications in all rock glacier inventories based solely on visual interpretation of aerial imagery, but the least amount of disagreement will undoubtedly be with our class 1 features. We believe our inventory can be compared, with minimal difficulty and high confidence, in international context to other rock glacier inventories that also identify "active" rock glaciers.*

[[Line 155: "after removing 146 small (< 0.01 km2) Class 3 rock glaciers following glaciological convention of area thresholds".
Why using this threshold size? Rock glaciers are not glaciers, neither are included in the World Glacier Inventory.]]

{{Rock glacier research is very much informed by glacier research, a discipline with much more robust inventories available due to much more robust remote sensing analysis techniques available, mostly stemming from the spectral reluctance of exposed ice. Glacier inventories have identified a lower area threshold beyond which glaciers cannot be confidently identified and delineated, and given rock glaciers are much more difficult to identify than glaciers it seems only prudent that we apply the same area threshold. In any event, the omitted features would have contributed only a minuscule fraction of the features included by count, and a virtually infinitesimal fraction of the features included by area.}}

*The question is not as straightforward as pictured by the authors. The WGI is largely based on satellite imagery, and glaciers are much more dynamic in nature than rock glaciers. As such, their mapping include uncertainty related to distinguishing between snow, ice and firn, which does not really apply to rock glaciers. Anyway, my question was*

*simple minded and I was just asking to write a sentence in which you justify your choice for such a threshold size.*

*While challenges relating to discriminating snow, ice and firm do not apply to rock glaciers, the challenges of identifying and delineating small rock glaciers are far greater, so using the same area threshold is prudent.*

[[Conclusions (lines 252-260): most of the conclusions paragraph reads more like introduction. Please consider rewriting and connecting the conclusions to the main results outlined in the manuscript.]]

{{Conclusions section in revised manuscript has been expanded and refocused as you suggest.}}

*Thank you for rewriting the conclusions.*

[[List of existing inventories:
I suggest adding the following references to the list of existing inventories, the first includes 5769 rock glaciers across Austria:
  Wagner et al 2020. The first consistent inventory of rock glaciers and their hydrological catchments of the austrian alps. Austrian Journal of Earth Sciences, 113: 1-23.
Brigitte Magori, Petru Urdea, Alexandru Onaca & Florina Ardelean (2020) Distribution and characteristics of rock glaciers in the Balkan Peninsula, Geografiska Annaler:
  Series A, Physical Geography, DOI: 10.1080/04353676.2020.1809905]]

{{Thank you for the suggestion, both references have been added to the revised manuscript.}}